# Uncovering population contributions to the extracellular potential in the mouse visual system using Laminar Population Analysis

**Atle E. Rimehaug**[1]*, **Anders M. Dale**[2], **Anton Arkhipov**[3], **Gaute T. Einevoll**[4,5]*

**1** Department of Informatics, University of Oslo, Oslo, Norway, **2** Department of Neuroscience, University of California San Diego, San Diego, California, United States of America, **3** Allen Institute, Seattle, Washington, United States of America, **4** Department of Physics, Norwegian University of Life Sciences, Ås, Norway, **5** Department of Physics, University of Oslo, Oslo, Norway

* atle.rimehaug@gmail.com (AER); gaute.einevoll@nmbu.no (GTE)

**Data Availability Statement:** The code for the LPA algorithm presented in this article as well as for the analysis of the results is publicly available at: https://github.com/atleer/Laminar-Population-

## Abstract

The local field potential (LFP), the low-frequency part of the extracellular potential, reflects transmembrane currents in the vicinity of the recording electrode. Thought mainly to stem from currents caused by synaptic input, it provides information about neural activity complementary to that of spikes, the output of neurons. However, the many neural sources contributing to the LFP, and likewise the derived current source density (CSD), can often make it challenging to interpret. Efforts to improve its interpretability have included the application of statistical decomposition tools like principal component analysis (PCA) and independent component analysis (ICA) to disentangle the contributions from different neural sources. However, their underlying assumptions of, respectively, orthogonality and statistical independence are not always valid for the various processes or pathways generating LFP. Here, we expand upon and validate a decomposition algorithm named Laminar Population Analysis (LPA), which is based on physiological rather than statistical assumptions. LPA utilizes the multiunit activity (MUA) and LFP jointly to uncover the contributions of different populations to the LFP. To perform the validation of LPA, we used data simulated with the large-scale, biophysically detailed model of mouse V1 developed by the Allen Institute. We find that LPA can identify laminar positions within V1 and the temporal profiles of laminar population firing rates from the MUA. We also find that LPA can estimate the salient current sinks and sources generated by feedforward input from the lateral geniculate nucleus (LGN), recurrent activity in V1, and feedback input from the lateromedial (LM) area of visual cortex. LPA identifies and distinguishes these contributions with a greater accuracy than the alternative statistical decomposition methods, PCA and ICA. The contributions from different cortical layers within V1 could however not be robustly separated and identified with LPA. This is likely due to substantial synchrony in population firing rates across layers, which may be reduced with other stimulus protocols in the future. Lastly, we also demonstrate the application of LPA on experimentally recorded MUA and LFP from 24 animals in the publicly available Visual Coding dataset. Our results suggest that LPA can be used both as a method to estimate positions of laminar populations and to uncover salient features in LFP/CSD contributions from different populations.

Analysis.git. The files necessary to run simulations of the model version used in the article as well as data resulting from simulations of that model version are publicly available in Dryad: Rimehaug, Atle et al. (2022). Uncovering circuit mechanisms of current sinks and sources with biophysical simulations of primary visual cortex [Dataset]. Dryad. https://doi.org/10.5061/dryad.k3j9kd5b8 The experimental data set utilized is publicly available at: https://portal.brain-map.org/explore/circuits/visual-coding-neuropixels.

**Funding:** This work was supported by the SUURPh program of Simula School of Research and Innovation (https://www.simula.no/), received by G.T.E., which provided a salary for A.E.R., the EU project EBRAINS 2.0 (Grant Agreement No. 101147319618), received by G.T.E., the National Institute of Neurological Disorders and Stroke of the National Institutes of Health (https://www.ninds.nih.gov/) under Award Number R01NS122742, by the National Institute of Biomedical Imaging and Bioengineering of the National Institutes of Health (https://www.nibib.nih.gov/) under Award Number R01EB029813, and by the Allen Institute, received by A. A. and providing a salary for A.A. The content is solely the responsibility of the authors and does not necessarily represent the official views of the National Institutes of Health. We acknowledge the use of Fenix Infrastructure resources, which are partially funded by the European Union's Horizon 2020 research and innovation program through the ICEI project under grant agreement No. 800858, received by G.T.E. None of the sponsors or funders had any role in the study design, data collection, analysis, decision to publish, or preparation of the manuscript.

**Competing interests:** The authors have declared that no competing interests exist.

## Author summary

To make the best use of all the data collected in neuroscientific experiments, we need to develop appropriate analysis tools. In extracellular electrophysiological recordings, that is, measurements of electrical signals outside of cells produced by neural activity, the low-frequency part of the signal referred to as the local field potential (LFP) is often difficult to interpret due to the many neurons and biophysical processes contributing to this signal. Statistical tools have been used to decompose the recorded LFP with the aim of disentangling contributions from different neural populations and pathways. However, these methods are based on assumptions that can be invalid for LFP in the structure of interest. In this study, we extend and validate a method called laminar population analysis (LPA), which is based on physiological rather than statistical assumptions. We tested, developed, and validated LPA using simulated data from a large-scale, biophysically detailed model of mouse primary visual cortex. We found that LPA is able to tease apart several of the most salient contributions from different external inputs as well as the total contribution from recurrent activity within the primary visual cortex. We also demonstrate the application of LPA on experimentally recorded LFP.

## Introduction

In recent years, there has been a prodigious increase both in the amount of data collected and the data made publicly available in neuroscience [1–5]. To make the most of the opportunities afforded by this wealth of data, it is imperative to develop analysis tools that help us make sense of the data and extract as much information about the underlying neural processes as possible.

In electrophysiological experiments, the extracellular potential can provide information both on the microscopic scale of single cells and on the mesoscopic scale of whole populations. The high-frequency part (above a few hundred Hz) of the potential is referred to as the multi-unit activity (MUA) and thought to mainly reflect the spiking activity of neurons in the vicinity of the electrode. The low-frequency part (below a few hundred Hz) of the potential is referred to as the local field potential (LFP) and thought to mainly reflect the transmembrane currents caused by input to populations of neurons near the electrode [6].

The applications of LFP have included investigations into sensory processing [7–15], motor planning [16, 17] navigation [18–22], and higher cognitive processing [23–27]. And the LFP is also a candidate signal for steering neuroprosthetic devices due to its relative stability over time compared to spikes [28–32]. However, for all its wide-ranging employment, the LFP is in many cases still not utilized or only partially utilized in analyses of electrophysiological data [6]. The reason is that the large number of neurons and diverse array of cell types and biophysical processes contributing to the LFP can often make it difficult to interpret [6, 33].

There have been important efforts to improve the interpretability of LFP. One of them has been to develop methods to calculate the current source density (CSD) from the LFP, which gives a more localized measure of activity that is easier to interpret in terms of the underlying neurophysiology [34–36]. The current sinks and sources in a CSD plot show where ions enter or leave neurons, and as such, it can provide information about where input arrives to a single cell or a population of cells. Despite being easier to evaluate than the LFP, however, the interpretation of current sinks and sources is also not straightforward. Current conservation dictates that any inward current must be balanced by an outward current, so a synaptic input

current is balanced by a stream of ions leaving the neurons [37]. The balancing outward current during synaptic input is the return current, which would show up as a current source in a CSD plot. However, an inhibitory synaptic input current will also produce a current source in a CSD plot, and there is typically no easy way to disambiguate the two potential origins [33, 38].

Biophysically detailed models of neurons have been used to simulate LFP in order to elucidate the mechanisms generating LFP, and by extension also the CSD [39]. This type of modeling has expanded our insight into the biophysical origins of extracellular potentials both on the single-cell [40–42] as well as the population level [43–53]. Unfortunately, constructing models that can simulate extracellular potentials can be a resource intensive endeavour, both with respect to human as well as computational resources. This is especially true for large-scale network modeling required to simulate the population LFP recorded in vivo [53]. It would therefore be highly advantageous to have analysis tools that allow for uncovering of biophysical processes from experimental data without requiring the construction of a model.

Among the methods that have been applied in pursuit of this aim are statistical decomposition tools such as independent component analysis (ICA) and principal component analysis (PCA). With these tools, the goal is to disentangle the LFP generated by the different pathways (or populations) contributing to the recorded LFP by utilizing statistical differences in their generation processes. ICA was successfully used to perform blind source separation on LFP recorded in hippocampus in [19, 54, 55], and [22] and in somatosensory cortex in [56]. Barth et al. (1989) established the use of PCA to decompose CSD by demonstrating its utility in rat barrel cortex [35, 57–59].

However, both in [58] and in [19] it is pointed out that the specific conditions that made PCA and ICA, respectively, appropriate in these structures and experimental or simulated conditions may not always be present [19, 58]. Both PCA and ICA rely on statistical assumptions —orthogonality in the case of PCA and statistical independence in the case of ICA—that may not be valid for the LFP generation processes in the structure of interest. Furthermore, the components they produce represent statistical features that are not immediately interpretable in terms of the populations or biophysical processes generating the LFP. Demonstrating this point, Głąbska et al. (2014) showed that when ICA was applied to LFP simulated with the Traub et al. (2005) model [60], they could only reliably uncover the contributions from two, at most three, populations by using knowledge about the populations from the model, which is information that's typically not available in experiments [61].

Seeking to address these limitations with the statistical tools for LFP decomposition, Einevoll et al. (2007) [62] developed an alternative method called laminar population analysis (LPA), which was based on physiological rather than statistical assumptions. This permits its use in structures and conditions where the statistical assumptions of ICA and PCA are invalid. LPA uses MUA and LFP obtained from the extracellular potential jointly in the decomposition. It begins by identifying laminar populations and the temporal profiles of their firing rates from the MUA data, and then uses the identified temporal profiles to decompose the LFP in the next step. The assumption here is that the observed LFP is caused by inputs generated by firing of action potentials in the presynaptic laminar populations identified from the MUA. The resulting components would then correspond to the LFP generated by each presynaptic laminar population, and thus, the components are directly interpretable in terms of underlying circuit dynamics.

The aim of this study is to further develop and validate the LPA method on simulated data from the large-scale biophysically detailed model of the mouse primary visual cortex developed by the Allen Institute [63]. We used the version presented in [53], which added feedback from the higher lateromedial visual area (LM) of the cortex to the feedforward input from LGN and

the background (BKG) Poisson source representing input from the rest of the brain, and was shown to reproduce not only experimentally observed spiking activity, but also LFP/CSD in response to a full-field flash stimulus. This model is unprecedented in its level of biological detail and recapitulated experimental observations in a data-driven manner, which makes it an ideal testbed for validation of the LPA method. We also compare the decomposition obtained with LPA with the ones obtained from applying PCA and ICA on the same data. Lastly, we demonstrate the use of the LPA method on the experimental data aquired with Neuropixel 1.0 probes in visual areas of mice that was made publicly available in [4].

The original formulation of the LPA method assumed that the LFP was predominantly generated by the recurrent activity within and between layers in the recorded structure. This can be true for anesthetized animals, where the communication between structures may be disrupted or the activity in some external areas is dampened more than others [64–66]. However, this is probably not always a valid assumption for awake animals [53]. Therefore, in this study, we extended the original formulation of the LPA method to take into account the contributions from external presynaptic populations in addition to the presynaptic populations that are internal to the structure in question.

We found that when applied to the simulated MUA data, the LPA method estimates the positions of layers in the model as well as the temporal profiles of their population firing rates well. We also find that it can disentangle the contributions from the different structures—LGN, recurrent activity from V1 as a whole, the feedback from LM, and the background input—with greater accuracy than the alternative, statistical decompositon methods PCA and ICA. Excessive synchrony in the firing rates of the different laminar populations impedes the separation of contributions from different layers within V1, but one can expect this to be ameliorated by the use of other experimental and simulated protocols in the future. We also show for an example animal that the LPA estimated position and timing of sinks and sources generated from feedback recapitulates their position and timing expected from modeling and anatomical data. The LPA algorithm and code to analyze the data utilized in this study are made freely available to the community at https://github.com/atleer/Laminar-Population-Analysis.git.

## Results

To develop and validate the LPA method, we utilized the large-scale, biophysically detailed model of the mouse primary visual cortex (area V1) developed by the Allen Institute [53, 63] (see Materials and methods). We used the version presented in [53], which was shown to reproduce salient features in experimentally recorded spikes and LFP/CSD simultaneously. Spikes and LFP across the laminar populations from an example simulation of a full-field white flash stimulus lasting 250 ms are displayed in Fig 1A and 1B. The raster plot is for a single trial, while the LFP is averaged over 10 trials. The model receives spikes recorded experimentally in the lateral geniculate nucleus (LGN) and the lateromedial (LM) area of visual cortex during presentations of the flash stimulus. The former provides a major feedforward visual input to V1, whereas the latter is a higher visual cortical area providing feedback input to V1. Additionally, the model also receives input from a Poisson source firing at a rate of 1 kHz, which represents the input from the rest of the brain. The model is composed of about 230,000 neurons, of which over 50,000 are biophysically detailed multi-compartment neurons that form a core of diameter 800μm and height 860μm. These neurons are positioned in five layers: layers 1, 2/3, 4, 5, and 6 (L1, L2/3, L4, L5, L6), where layers 2 and 3 have been merged into a single layer. Each layer except L1 has a single class of excitatory neurons and three classes of inhibitory neurons (Pvalb, Sst, and Htr3a). L1 only has one inhibitory population (Htr3a) and no excitatory cells.

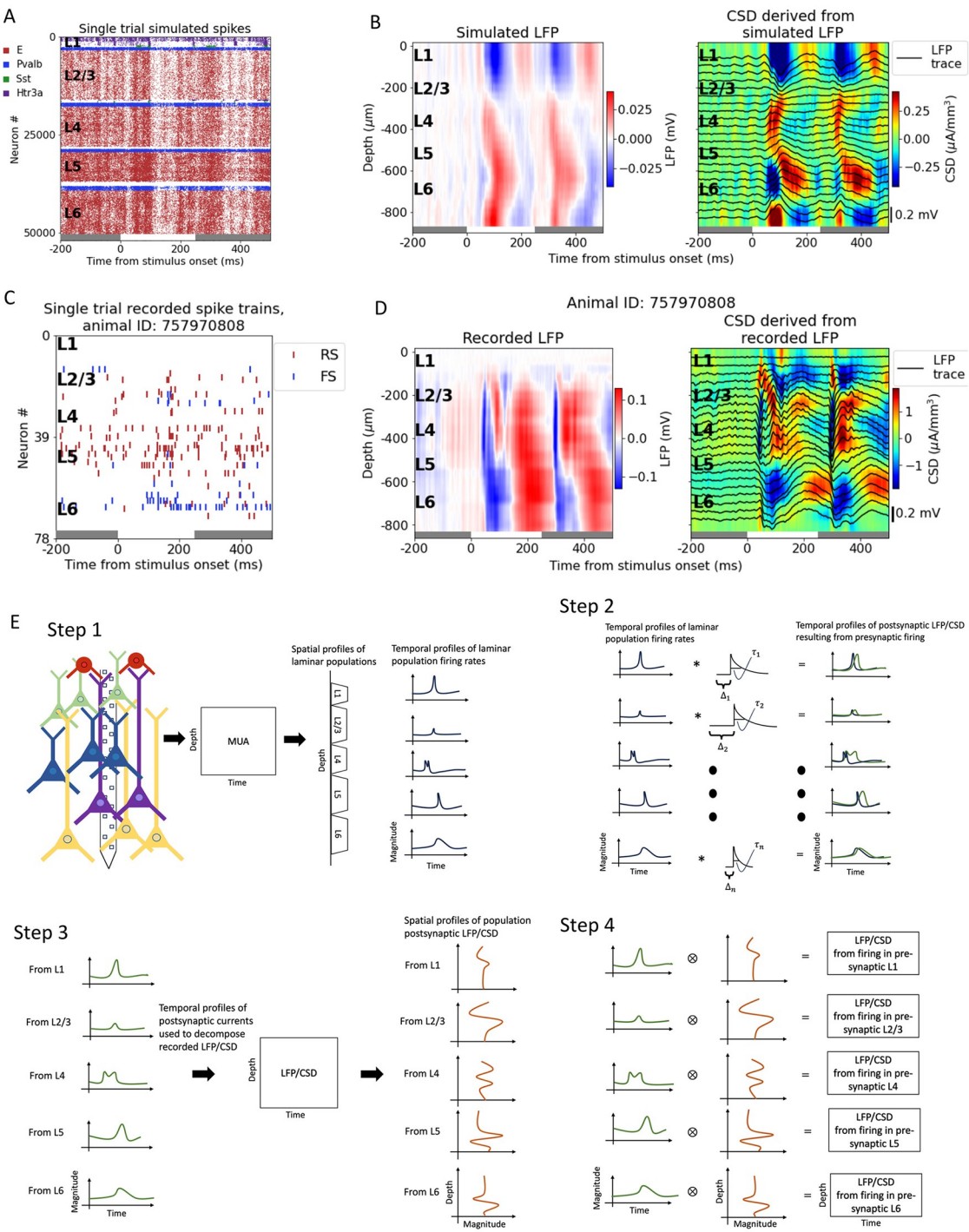

**Fig 1. Illustrating the LPA method and the data sets.** (A) Raster plot of spikes from a single trial simulation of a full-field white flash stimulus. The 250 ms where the stimulus is presented is marked with a horizontal white bar between gray bars at the bottom of the plot. Spikes from excitatory cells are red, from Pvalb-cells blue, from Sst-cells green, and from Htr3a-cells purple. (B) Raster plot of spikes from a single trial of a full-field white flash recorded in animal. Spikes from regular-spiking cells are shown in red and spikes from fast-spiking cells in blue. (C) LFP (left) and CSD (right) calculated from the LFP from a simulation of a full-field white flash stimulus to an example animal in the experimental dataset. Averaged over 10 trials. (D) LFP (left) and CSD (right) calculated from the LFP recorded during presentations of a full-field white flash stimulus. Averaged over 75 trials. (E) Schematic of the steps of the LPA method as described in the main text.

We also demonstrate the use of the LPA method on experimental recordings from mouse primary visual cortex in the publicly available Visual Coding dataset collected at the Allen Institute [4] (see Materials and methods). Six Neuropixel 1.0 probes recorded the extracellular potential across multiple brain areas, with a focus on six cortical (V1, LM, AL, RL, AM, PM) and two thalamic (LGN, LP) visual areas. Spikes from a single trial and LFP averaged over 75 trials recorded from V1 in an example animal are displayed in Fig 1C and 1D. The CSD derived from the simulated and the recorded LFP is shown to the right of the LFP plots in Fig 1B and 1D, respectively. The CSD provides a more localized measure of current sinks and sources that can be easier to interpret than the LFP, and was calculated using the delta iCSD method (see Materials and methods) [6, 36].

The LPA method can be divided into four steps (Fig 1E). In the first step, the recorded MUA is decomposed into spatial and temporal profiles of components corresponding to the laminar populations in the structure. The number of populations is assumed a priori. In Fig 1E, the number of populations has been set to five, equal to the number of distinct layers in the V1 model. The spatial profile of the components obtained from the MUA indicate the position of the different layers. The temporal profiles show the time course of the firing rates of these LPA-identified laminar populations. In the second step, the temporal profiles of the laminar firing rates are convolved with kernels (here illustrated as exponential functions) to produce the time course of the postsynaptic LFP. Convolving with these kernels represents the process by which action potentials in the presynaptic population results in the postsynaptic LFP. These postsynaptic temporal profiles are then used in the third step to decompose the recorded LFP (or CSD) to obtain the spatial profiles of the LFP/CSD generated by each presynaptic population. In the fourth and final step, the temporal and spatial profiles of the postsynaptic LFP/CSD are multiplied to produce the LFP/CSD across the recorded depth resulting from spikes firing in each presynaptic laminar population identified in step 1. These population contributions to the LFP/CSD are then summed to obtain the total LFP/CSD estimated with LPA and compared to the original, recorded LFP/CSD. Steps 2 to 4 are repeated with varying time constants $\tau$ and delays $\Delta$ for the kernels until the discrepancy between the total LFP/CSD estimated from summing the LPA-estimated population contributions and the recorded LFP/CSD is minimized.

## LPA can identify layer positions and laminar firing rates

We tested the first step (Fig 1E) of the LPA algorithm by applying it to the MUA obtained from a simulation of two different types of full-field stimuli: a white and a black screen flash (Fig 2). Each stimulus type is presented for 250 ms in 10 trials, with 500 ms of gray screen between each presentation. The purpose of applying LPA to the MUA is to identify the positions of populations in a structure and to estimate the temporal profile of their population firing rates. In the optimization procedure, the total MUA calculated from the estimated population positions and temporal profiles is compared to the total, original MUA to which LPA is applied. The procedure is stopped when the difference between the original MUA and the MUA calculated from the LPA-estimated positions and temporal profiles is minimized. We here report the correlation and relative mean square error between the original, simulated MUA and the MUA calculated from the final, optimized LPA-estimate. Assuming five populations (equal to the number of layers in the model), the relative mean square error of the LPA-estimates are 0.099 and 0.101 for the white and black flash, respectively, while the correlations between the LPA-estimated and the simulated MUA are 0.76 and 0.79 for the same stimuli (Fig 2A). The spatial profiles of the five components cover each of the five layers from L1 to L6 (Fig 2B and 2C). Here, the layer that the spatial profile of a component overlaps the most with

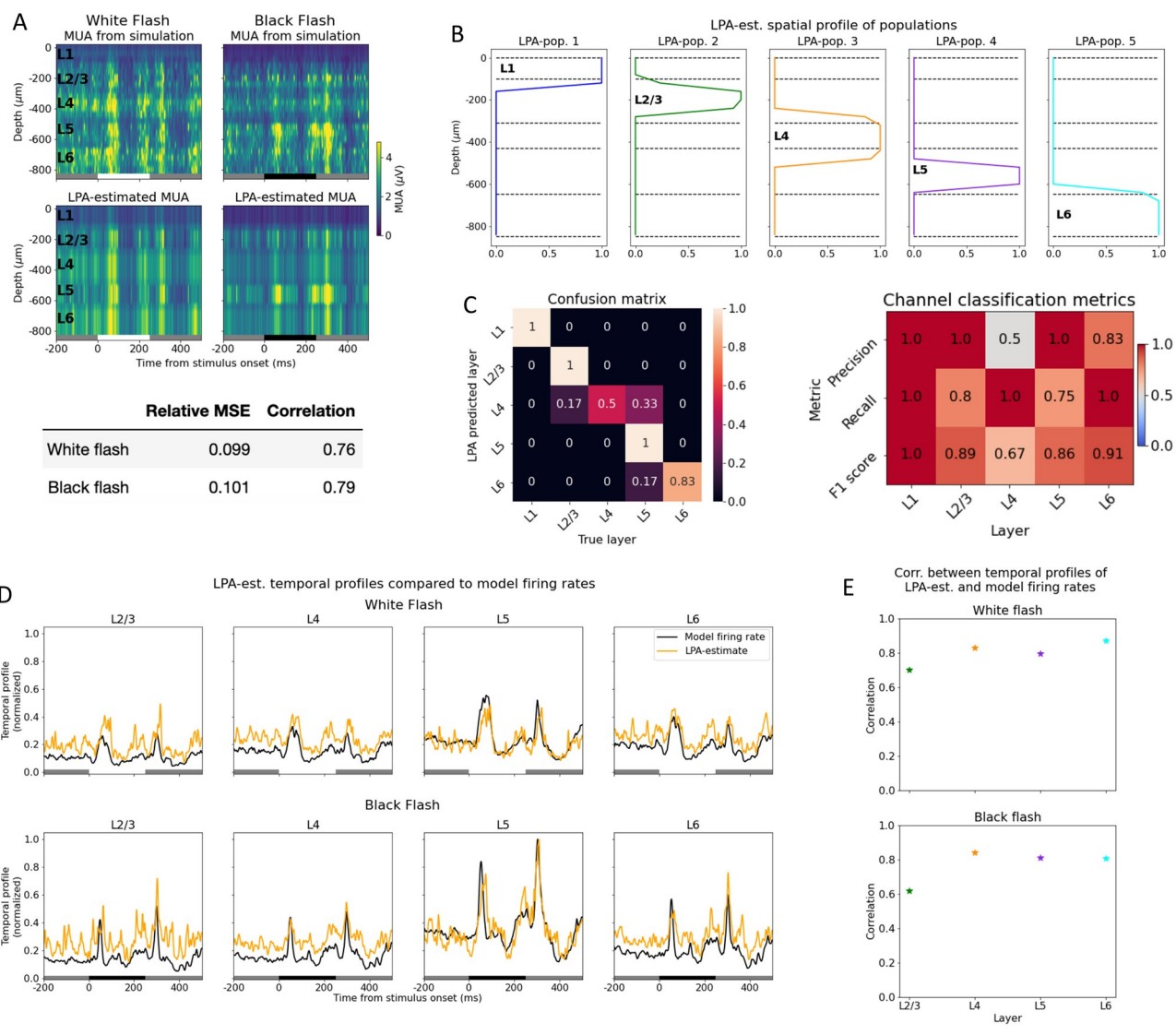

**Fig 2. LPA can identify layer positions and laminar firing rates.** (A) Top: Trial-averaged MUA from simulation with white (left) and (black) flash stimuli and MUA estimated from LPA-components below. Averaged over 10 trials. White and black bars at the bottom of the plots indicate when the white and black flash, respectively, is presented. Bottom: Relative MSE and correlation between MUA from simulation and MUA estimated from LPA-components (B) Spatial profiles of LPA-components along the axis of simulated recording probe. (C) Left: Confusion matrix for classification of electrode position from spatial profiles in (B). Right: Precision, recall, and F1 score for predicted layer position of electrodes by LPA. (D) Temporal profiles of LPA-components together with temporal profiles of laminar firing rates of excitatory cells in the model. Temporal profiles of both LPA-components and the model firing rates are normalized to the highest firing rate across all populations and both stimuli. (E) Correlation between temporal profiles of LPA-components and laminar firing rates in (D).

is considered to be the layer that this component represents. In other words, the spatial profiles of the components delineate the position of each layer estimated by LPA.

The identification of laminar positions by LPA is a multiclass classification problem. Therefore, we computed the confusion matrix and the associated precision, recall, and F1 scores, which are frequently used metrics in the assessment of classification algorithms [67–69] (see Materials and methods). What is quantified with these metrics is whether the layer position of each recording electrode determined by LPA corresponds to the true layer position of that electrode. The LPA algorithm predicts the location of the layers, but the predicted location can

only be assessed at the position of the electrode contacts as that is where we have data points. In the confusion matrix, the layers predicted by LPA are on the y-axis and the true layer positions of electrodes are on the x-axis. Each element is normalized by the total number of channels predicted by LPA to be in a given layer, i.e., by the sum in each row. A true positive (TP) classification means that an electrode predicted to be in a particular layer (say, L4) is in fact located in that layer, and the true positives lie along the diagonal of the confusion matrix. A false positive (FP) means that an electrode predicted to be located in a particular layer is in fact in a different layer. True negative (TN) means that an electrode predicted to be in a different layer by LPA is in fact located in a different layer. A false negative (FN) means that an electrode predicted to be located in a different layer is in fact in the layer in question.

For all metrics utilized, scores closer to 1 indicate higher correspondence between the true and LPA-predicted laminar positions of probe channels, while scores closer to 0 indicate a lower correspondence. Precision is calculated by dividing the number of true positives by the sum of the number of true positives and the number of false positives: $precision = \frac{TP}{TP+FP}$ [69]. Precision assesses the extent of type 1 errors (false positives) made in the classification; the closer the precision is to one, the fewer type 1 errors are made. Recall is calculated by dividing the number of true positives by the sum of true positives and false negatives: $recall = \frac{TP}{TP+FN}$. Recall assesses the extent of type 2 errors (false negatives) made in the classification. F1 score is the harmonic mean of precision and recall and is computed as $F_1 = \frac{2 \times precision \times recall}{precision+recall}$. The F1 score is thus an aggregate metric of the precision and recall and provides a balanced measure of the extent of type 1 and type 2 errors.

The precision of the LPA classification is 1 for all layers except L4 and L6, where it is 0.5 and 0.83, respectively (Fig 2C, right). This means that for L1, L2/3, and L5, there are no false positives and all the positive LPA classifications for these layers are correct. For L6, the classifications are predominantly true positives, but there are also some false positives, while for L4, half of the positive classifications are true positives and half are false positives. This aligns with what we can infer from a visual inspection of the spatial profiles shown in Fig 2B, where the profile of the third LPA-population seems to bleed over into the neighboring L2/3 and L5, even though it overlaps most with L4. The fifth population, which mainly covers L6, also covers a part of L5. The recall of the LPA classification is 1 for L1, L4, and L6, while it is 0.8 and 0.75 for L2/3 and L5, respectively. This means that for L1, L4, and L6, no electrodes are incorrectly predicted to be located in a different layer (i.e., no false negatives). For L2/3 and L5, some electrodes are falsely predicted to be located in other layers by LPA, but the majority of the true positives are correctly identified. The F1 score combines these two metrics and is 1 for L1, 0.89 for L2/3, 0.67 for L4, 0.86 for L5, and 0.91 for L6, showing that the estimate is best for L1 and worst for L4.

The temporal profiles of the laminar population firing rates estimated with LPA together with the true laminar population firing rates are shown in Fig 2D. The firing rates of only excitatory cells were used as the benchmark for all populations because they account for the 88.5% of the activity in the recorded MUA (see Supplementary S1 Fig), and thus, the temporal profiles predicted by LPA can be expected to predominantly reflect the firing rates of excitatory cells. L1 was not included for this part of the analysis since there are no excitatory cells in that layer in the model. We computed the correlation between the LPA-estimated and the true laminar firing rates in response to the white and the black flash stimuli (Fig 2E). For the white flash, the correlations were: L2/3: 0.70, L4: 0.83, L5: 0.80, L6: 0.87, while for the black flash, the correlations were: L2/3: 0.61, L4: 0.84; L5: 0.81, L6: 0.81. Thus, correlations were 0.7 or higher for all populations for both flashes, except L2/3 during black flash presentation, which indicates that the main features of the firing rates—the timing of the peaks after stimulus onset and

offset and the magnitudes of the peaks relative to baseline—are captured reasonably well by the temporal profiles estimated with LPA.

These results demonstrate that LPA can be used as a method to estimate the positions of layers in electrophysiological recordings either as a supplement to histological information or as a substitute when histological data is unavailable. They also show that an estimate of the temporal profile of the laminar population firing rates can be obtained from the MUA with the LPA method.

## Excessive synchrony impedes the separation of laminar LFP/CSD contributions

After the laminar positions and the corresponding temporal profiles of laminar firing rates had been identified from the MUA, these temporal profiles were utilized in the decomposition of the recorded LFP/CSD [62] (steps 2 and 3 in Fig 1E). As noted above, the temporal profiles estimated from MUA will likely predominately reflect firing of excitatory neurons. However, unless excitatory and inhibitory cells happen to be spatially separated, which they are not here, their contributions to the MUA cannot be separated and there will likely be minor contributions from inhibitory spikes in the temporal profiles estimated for each population. Spikes from excitatory and inhibitory neurons will generate currents with opposite signs in the post-synaptic cells; excitatory synaptic input will generate a current sink while inhibitory synaptic input will generate a current source. These opposite effects will not be captured when a single temporal profile is used for a whole laminar population that contains both inhibitory and excitatory neurons. However, since the MUA generated from excitatory spikes make up the vast majority of the total MUA (88.5%; see Supplementary S1 Fig), the error from this simplification can be expected to be minor.

We first convolved the population temporal profiles with suitable kernels to get the temporal profiles of the postsynaptic CSD resulting from firing of action potentials in the presynaptic population. Here, we used an exponential function for the kernels because it produced the best correspondence between the total CSD estimated by LPA and the original simulated CSD, but other kernel shapes can also be used [62]. The delays and time constants of the kernels are unique to each population, and these parameters are iteratively optimized.

In the original formulation of the LPA method, the LFP generated by input from external structures (such as LGN, LP or higher cortical areas in the case of V1) was assumed to be negligible relative to the LFP generated from recurrent activity within the structure [62]. This may in part hold true for some areas in experiments carried out on anesthetized animals, where the anesthesia affects external structures more than the structure of interest and/or disrupts communication between structures. For example, it has been shown that activity in structures external to V1, such as LGN and the latermedial (LM) area of the higher visual cortex, is dampened more by anesthesia than activity in V1 [64–66]. Thus, contributions from external structures may be strongly reduced under these experimental conditions. However, for experiments carried out on awake animals, the LFP arising from input from external structures can be significant and this assumption would then be invalid [53, 66, 70]. Therefore, we extended the decomposition of the LFP in the LPA method such that contributions from external structures could be included. This was done by simply appending the temporal profiles of the firing rates of the external structures to the temporal profiles of the laminar population firing rates, and convolving the temporal profiles of the external structures with kernels to get the postsynaptic LFPs they generate.

In the V1 model version utilized here, the model receives input from LGN, the higher visual cortical area LM, and a Poisson source firing at 1 kHz [53]. Both the LGN input and the input

from LM were generated from spikes recorded experimentally in those structures during presentations of the full-field flash stimuli. With the extended version of LPA, that results in temporal profiles from seven populations for this model: four laminar populations with excitatory cells (L2/3 to L6) identified from the MUA and three external structures (Fig 3A).

The trial-averaged CSD derived from the simulated LFP in response to white and black flashes is displayed in the panels at the top of Fig 3B, and the total CSD estimated from the components obtained after applying the LPA decomposition to this simulated CSD is shown in the panels below. The relative MSE between the simulated and the LPA-estimated total CSD was 0.13 for the white flash and 0.19 for the black flash, while the correlation was 0.94 and 0.92 for the same flashes, respectively (Fig 3B, bottom), which indicates that the estimate from the LPA components captures both the magnitude and the shape and timing of the main sinks and sources in the simulated CSD. The spatial profiles for the CSD from each population were obtained according to Eq 6 by using the simulated CSD and the temporal profiles (see Materials and methods) (Fig 3C). Then, these spatial and temporal profiles of the CSD were multiplied to get the CSD produced by the firing in each presynaptic population (Fig 3D).

These LPA-estimated population contributions to the CSD were then compared to the true CSD contributions (see Materials and methods) produced by each presynaptic population in the model (Fig 3E). By visual inspection, it appears that the position and timing of some salient sinks and sources are captured for LGN and the feedback (FB). There are early sinks approximately in L4 and in L5/L6 in the contribution from LGN firing (compare the upper left panels of Fig 3D and 3E) arising around 50 ms after flash onset, which are subsequently followed by sources in the same regions plus a sink covering L1 and parts of L2/3. This pattern repeats 50 ms after flash offset at 250 ms. The LPA-estimated CSD from feedback exhibits an upper layer dipole with a sink at the top and a source below it after both flash onset and offset, which recapitulates the distributions of sinks and sources in the true CSD contribution from feedback. However, the magnitude of the dipole is greater in the LPA-estimate than in the true CSD. Lastly, LPA also captures the relative weakness of the CSD generated by the background (BKG).

The results are more variable for the LPA-estimated CSD contributions from the laminar populations of V1, however. For L4 and L6, some of the deep layer sinks and sources appear to be captured, though not in the right magnitude for L6. While for L2/3 and L5, there seems to be little correspondence to the true CSD in the timing and position of sinks and sources. The relative MSE and the correlation between the LPA-estimated and the true CSD for the above-mentioned populations support this analysis (Fig 3D, bottom). The correlation is 0.54 or higher for all the external structures, and the relative MSE is 0.6 for LGN, and 0.7 for the FB, and 1.4 for the BKG. The correlation is 0.44 for L4 and 0.58 for L6 and the relative MSE is 0.8 for L4 and 2.7 for L6. The higher relative MSE for L6 likely reflects the deviation in magnitude for the salient sinks and sources. For L2/3 and L5, however, the correlation is 0.01 and -0.07, respectively.

The relative inability of LPA to correctly characterize the contributions from recurrent activity for some populations within V1 compared to the contribution from external input is likely caused by the high synchrony between the firing rates of the laminar populations in V1 (Fig 3G). The correlation between the population firing rates from L2/3 to L6 throughout the simulations of both flashes are all above 0.83, while the correlation between the external structures LGN and FB is 0.62, and the correlation between BKG and LGN and FB is -0.02 and 0.0, respectively. When the activation of various presynaptic populations is highly synchronous, it can result in significant cross-contamination in the CSD attributed to these different populations. This has previously been observed for other decomposition methods that rely on

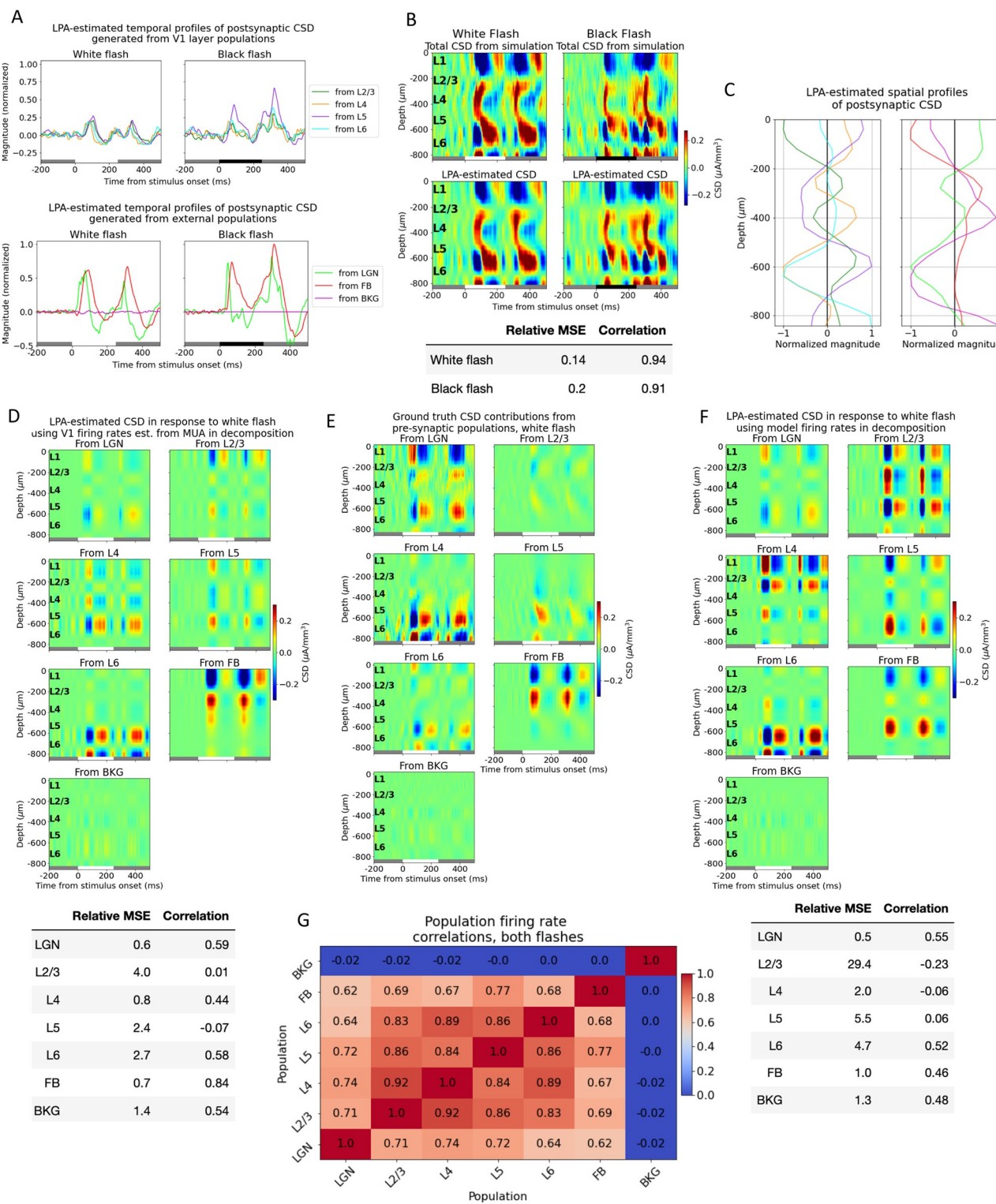

**Fig 3. Excessive synchrony impedes the separation of laminar LFP/CSD contributions.** (A) Temporal profiles of postsynaptic CSD of each LPA-component in response to white and black flash. White and black bars at the bottom of the plots indicate the stimulus periods. (B) Top: CSD from simulation with white (left) and (black) full-field flash stimuli and CSD estimated from LPA-components below. Bottom: Relative MSE and correlation between CSD from simulation and CSD estimated from LPA-components (C) Spatial profiles of LPA-components along axis of simulated recording probe. (D) Top: White flash CSD generated from firing in each presynaptic population estimated with LPA using temporal profiles obtained from MUA as shown in Fig 2D. Bottom: Relative MSE and correlation between LPA-estimated and true white flash CSD contributions from each presynaptic

population. (E) True white flash CSD contributions from firing in each presynaptic population. (F) Top: White flash CSD generated from firing in each presynaptic population estimated with LPA using temporal profiles obtained from firing rates of excitatory cells in each layer from 2/3 to 6 in the model. Bottom: Relative MSE and correlation between LPA-estimated and true white flash CSD contributions from each presynaptic population. (G) Correlation between firing rates of different populations, both flashes together.

differences in temporal activation, such as independent component analysis (ICA) [19], and is likely to affect LPA decomposition as well.

Up until this point, we utilized temporal profiles of the population firing rates estimated from the MUA for the V1 populations. However, for the external populations, we used the temporal profiles of the population firing rates calculated directly from the spikes in these populations. We do not simulate the ECP for the structures providing external input to the model —the LGN, LM, and background. Therefore, the temporal profile of the firing rates in these structures cannot be estimated from MUA in the model and must be calculated from the spikes in the input. The magnitudes of the temporal profiles of firing rates estimated from MUA in V1 are not directly comparable to the magnitudes of the firing rates calculated from spikes in the external structures. Therefore, it is not clear whether the temporal profiles estimated from MUA in V1 can be combined directly with the temporal profiles computed from spikes in the external populations in the decomposition of the CSD; it may result in a misestimation of the LFP contribution from V1 relative to that generated by the external input. It could be that deviations from the true magnitude of firing in the estimate from MUA will simply be compensated by the magnitudes of the spatial profiles, such that the magnitudes of the population MUA estimated with LPA are still correct, but that cannot be guaranteed. This potential issue of lack of compatibility in firing rate estimates utilized in the LFP decomposition does not affect the application of LPA to experimental data, as the temporal profiles of firing rates will be estimated from recorded MUA in all structures in experiments.

Therefore, to eliminate potential firing rate compatibility issues as a potential source of error when applying LPA to the model, we decided to employ the true firing rates calculated for the excitatory cells in the V1 model instead of the temporal profiles estimated from MUA in the decomposition of the CSD. We only used spikes from excitatory cells because the total MUA primarily reflects firing of excitatory populations in the model (S1 Fig) and it was shown in [53] that the main sinks and sources in the CSD stem primarily from excitatory input in this model. However, this did not improve the correspondence between the LPA estimate and the true CSD contributions for the laminar populations (Fig 3F). The correspondence was still best between the predicted CSD and the CSD generated from external structures, while the contributions from different populations within V1 were not well separated and identified (with the partial exception of L6, where the correlation was 0.52 but the relative MSE was still high at 4.7).

## LPA identifies salient sinks and sources from V1 as a whole

Since the CSD contributions from some of the different laminar populations were not well separated, we explored whether the contribution from different populations would be better captured with fewer populations assumed in the LPA decomposition. The results from applying LPA with two populations assumed are shown in S11 Fig. We merged the firing rates of L2/3 and L4 into one upper layer population firing rate and the firing rates of L5 and L6 into one deep layer population firing rate. The firing rates from external structures were unchanged. With this configuration, the contribution from the upper layer population was reasonably well estimated for the white flash response, with a correlation to the true CSD from these layers of 0.64 and a relative MSE of 1.2 (S11D Fig). The LPA-estimated CSD contributions from

external structures were similar to their estimates with all layers distinguished. However, the deep layer contribution was not well estimated, with a correlation to the true CSD of 0.14 and 0.35 and a relative MSE of 2.1 and 7.4 for the white and black flash response, respectively (S11D and S11E Fig).

We experimented with other numbers of populations for V1, but achieved the best results when we merged all the laminar populations of V1 together into one single population. We continued to use the firing rates of the excitatory cells of the model rather than the temporal profiles estimated from the MUA in the decomposition of the CSD to circumvent the issue of how to combine temporal profiles of firing rates estimated from the MUA of V1 with the firing rates calculated from spikes in the external structures. However, the results from using temporal profiles estimated from MUA rather than firing rates of excitatory cells in the model in the decomposition can be found in the supplementary figures S5 and S6 Figs.

Thus, the firing rates of four presynaptic populations—LGN, V1, FB, and BKG—were now convolved with kernels to get the temporal profiles of the postsynaptic CSD and used in the decomposition of the simulated CSD (Fig 4A–4C). The relative MSE between the total CSD estimated from the LPA components and the simulated CSD was 0.19 and 0.26 for the white and the black flash, respectively. The correlations were 0.93 and 0.89 for the same flashes. This is a slight increase in the relative MSE (from 0.13 and 0.19) and a slight decrease in the correlations (from 0.94 and 0.92) from the LPA estimate with the five laminar populations in V1 distinguished. A somewhat worse fit to the total CSD is to be expected since there are fewer parameters to optimize. However, these values and visual comparison both indicate that the LPA-estimate still captures both the magnitudes and the distribution of sinks and sources in the total simulated CSD. The resulting spatial profiles of the population CSD (Fig 4D) were multiplied with the temporal profiles (Fig 4B) to produce the estimated CSD contributions from firing in each presynaptic population (Fig 4E, top).

Comparing the estimated and the true CSD contribution from V1 as a whole visually, it appears that the prominent deep layer (in L5/L6) oscillations of sinks and sources arising around 50 ms after flash onset and offset are recapitulated by the LPA estimate. The relative MSE between the LPA estimate and the true CSD has decreased significantly and the correlation increased significantly relative to the LPA estimates with the different laminar populations distinguished (see Figs 4F and 3D and 3F, bottom). The relative MSE increased from 0.5 to 1.1 and the correlation decreased from 0.55 to 0.22 for LGN, possibly owing to the sinks and sources that have appeared at the deepest channels in this estimate, which aren't present in the true CSD from LGN. For the FB estimate, the relative MSE increased from 1.0 to 1.7 and the correlation also increased from 0.46 to 0.62, but the FB estimate still exhibits the spurious L5/L6 source not seen in the true CSD. The relative weakness of the contribution from the background input is still largely recapitulated, but the relative MSE has increased. And comparing the estimated and true CSD from BKG visually, we see that the magnitude of the estimate has increased somewhat. Overall, even though there are still some significant sinks/sources in the LPA estimate not observed in the true CSD, the LPA estimate did improve by merging the different laminar populations of V1 into one single population. Thus, from here on, we only used one population for V1 in the decomposition.

## Penalizing deviations from zero in the summed CSD across channels improves the LPA prediction

Multiple methods have been developed to calculate the CSD from the recorded LFP [36]. The traditional and simplest method is to multiply the discrete double spatial derivative of the LFP with the negative conductivity: $C = -\sigma \frac{\delta \phi_L}{\delta z}$, where $C$ is the CSD, $\phi_L$ is the LFP, $\sigma$ is the

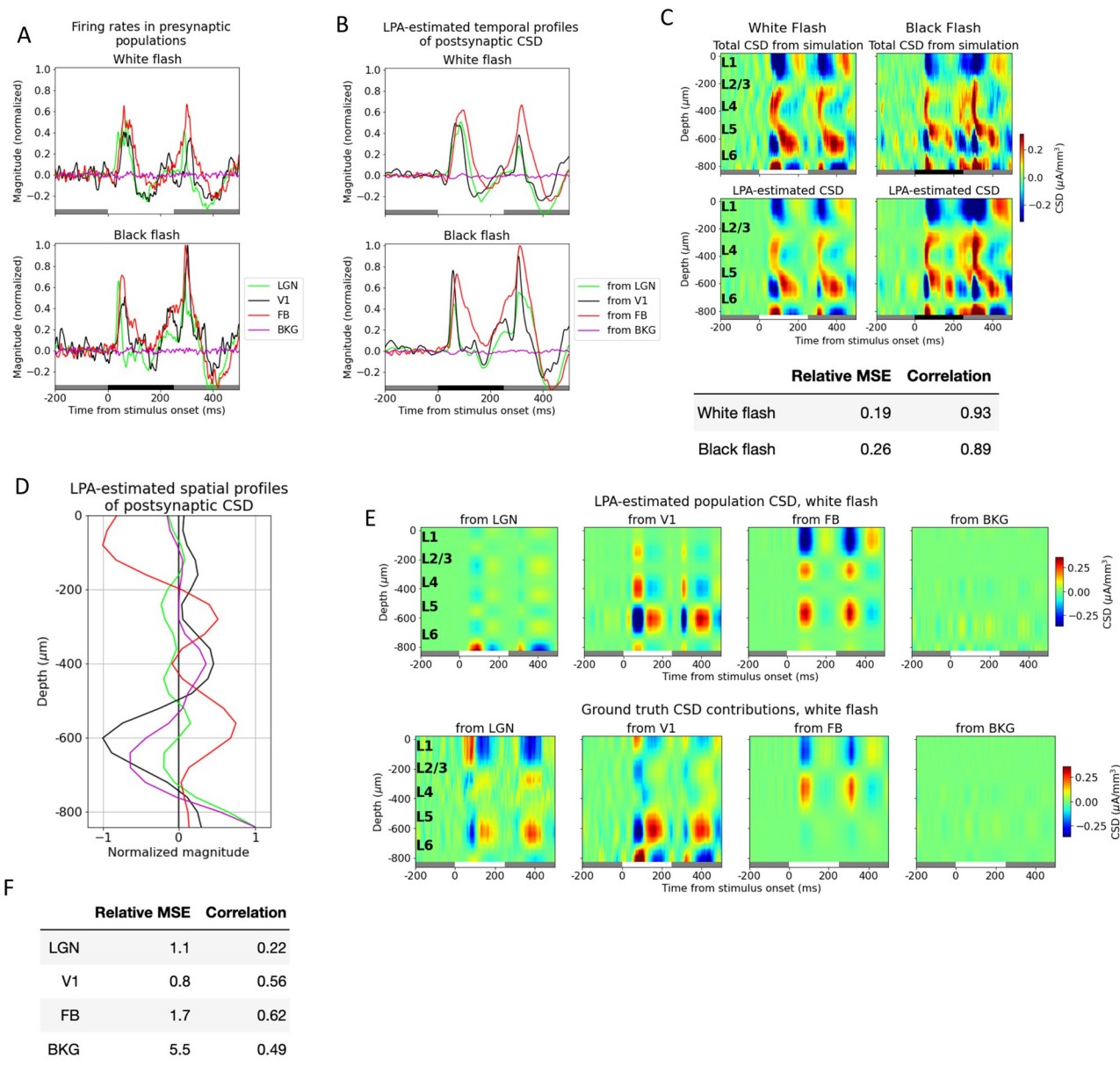

**Fig 4. LPA identifies salient sinks and sources from V1 as a whole.** (A) Temporal profiles of firing rates of presynaptic external structures (LGN: green line; feedback: red line; background: purple line) and excitatory cells in V1 (black line) where the layer populations of V1 have been merged into one population. White and black bars at the bottom of plots indicate when the stimulus is presented. (B) Top: Trial-averaged CSD from simulation with white (left) and (black) flash stimuli and CSD estimated from LPA-components using presynaptic firing rates in (A) in decomposition. Bottom: Relative MSE and correlation between CSD from simulation and CSD estimated from LPA-components. (C) Spatial profiles along axis of simulated recording probe of LPA-components obtained using populations in (A) in decomposition of the simulated CSD. (D) Temporal profiles of postsynaptic CSD of each LPA-component. (E) Top: LPA-estimated CSD generated from firing in each presynaptic population shown in (A). Bottom: True CSD generated from firing in the same presynaptic populations. (F) Relative MSE and correlation between LPA-estimated CSD and true CSD generated from each presynaptic population.

conductivity, and $z$ is the depth along the probe axis. In this method, the CSD is assumed to be constant in the infinite plane orthogonal to the probe axis. This assumption is generally not valid, and can lead to significant deviations in the amplitude of the CSD estimate from the true amplitude [36, 43]. One method to calculate the CSD that does not assume infinitely constant CSD in the plane is the delta iCSD method, where the CSD is only assumed to be constant

within a certain radius in the plane [36]. The V1 model utilized here has a radius of 400μm, and for this reason, we utilized the delta iCSD method with the CSD assumed to be constant within a radius of 400μm around the electrode in our calculation of the CSD.

The total amount of current entering or leaving the extracellular space from an individual neuron has to sum to zero according to Kirchoff's current law. If the CSD is calculated with the traditional method, the current sinks and sources summed across electrodes will by design sum to zero [36]. If the CSD is calculated with the delta iCSD method, however, the CSD summed across electrodes may deviate from zero [36]. This may partly explain why we observe a spurious source in L5/L6 of the FB-contribution estimated from LPA, while no such source is present in the true CSD from FB. Based on this observation, we implemented a penalty on deviations from zero in the sum of CSD across depths. This was done by adding a penalty term $\lambda \frac{1}{B} \sum_j^B \ \sum_i^{N_{ch}} \phi^C(z_i, t_j)$ to the cost function (Eq 8, Materials and methods), where the $\lambda$ term sets the magnitude of the penalty.

We varied the magnitude of the penalty and observed its effect on the LPA-prediction. We found that with no penalty, the deviation from zero was 0.082 in the total CSD estimated from the LPA components and 0.047 for the original simulated CSD (Fig 5A), where the deviation is normalized to the maximum value of the CSD. The deviation from zero in the LPA-estimated CSD decreased with increasing magnitude of the penalty control parameter $\lambda$ until $\lambda = 4$, at which point the deviation plateaued (with some variability around the value of the deviation for the original simulated CSD). The deviation from zero for the LPA-estimated CSD at $\lambda = 4$ was 0.045. The relative MSE between the total LPA-estimated CSD and the total simulated CSD increased with increasing penalty, while the correlation decreased (Fig 5B). The total CSD estimated with LPA for $\lambda = 4$ is shown together with the original simulated CSD for both flashes in Fig 5C. Even though the error of the estimate is greater than it was without the penalty on deviations from zero (see Figs 3C and 4B), the salient features are still reproduced, with sinks and sources appearing about 50 ms after both flash onset and offset at the same depths as in the original simulated CSD.

Despite the increase in the error for the total CSD estimate with the added penalty, the estimates of the population contributions to the CSD were improved (Fig 5D). And the estimates of the population contributions to the CSD were optimal for the values at which the deviation from zero was smallest. The smallest deviation was observed at $\lambda = 4$, and we therefore used the LPA-estimate obtained for this value in Fig 5, but the results were similar for the neighboring values ($\lambda = 3$ to $\lambda = 6$), which also produced small deviations. Comparing the feedback estimate at $\lambda = 4$ with the ground truth (Fig 5E), we can observe that the spurious source in L5/L6 is still present, but has diminished in magnitude relative to the estimate with no penalty on deviation from zero (Fig 4E). The relative MSE went from 1.1 to 0.6 and the correlation increased from 0.22 to 0.65 for the LGN estimate (Fig 5F). For the V1 estimate the relative MSE went from 0.8 to 0.6 and the correlation from 0.56 to 0.67, while for the feedback the relative MSE decreased from 1.7 to 0.9 and the correlation increased from 0.62 to 0.75. The relative MSE decreased from 5.5 to 2.6 and the correlation also decreased 0.49 to 0.32 for the background. Thus, with the exception of the correlation for the background, the estimates improved for all populations on both metrics.

## Comparison with ICA and PCA decomposition

For comparative purposes, we applied the alternative, statistical spatiotemporal decomposition methods ICA and PCA to the same CSD data (S9 and S10 Figs). The direct, physiological interpretation of the components afforded by LPA is not readily available for the components obtained from ICA or PCA. This does not mean that the components cannot be interpreted or

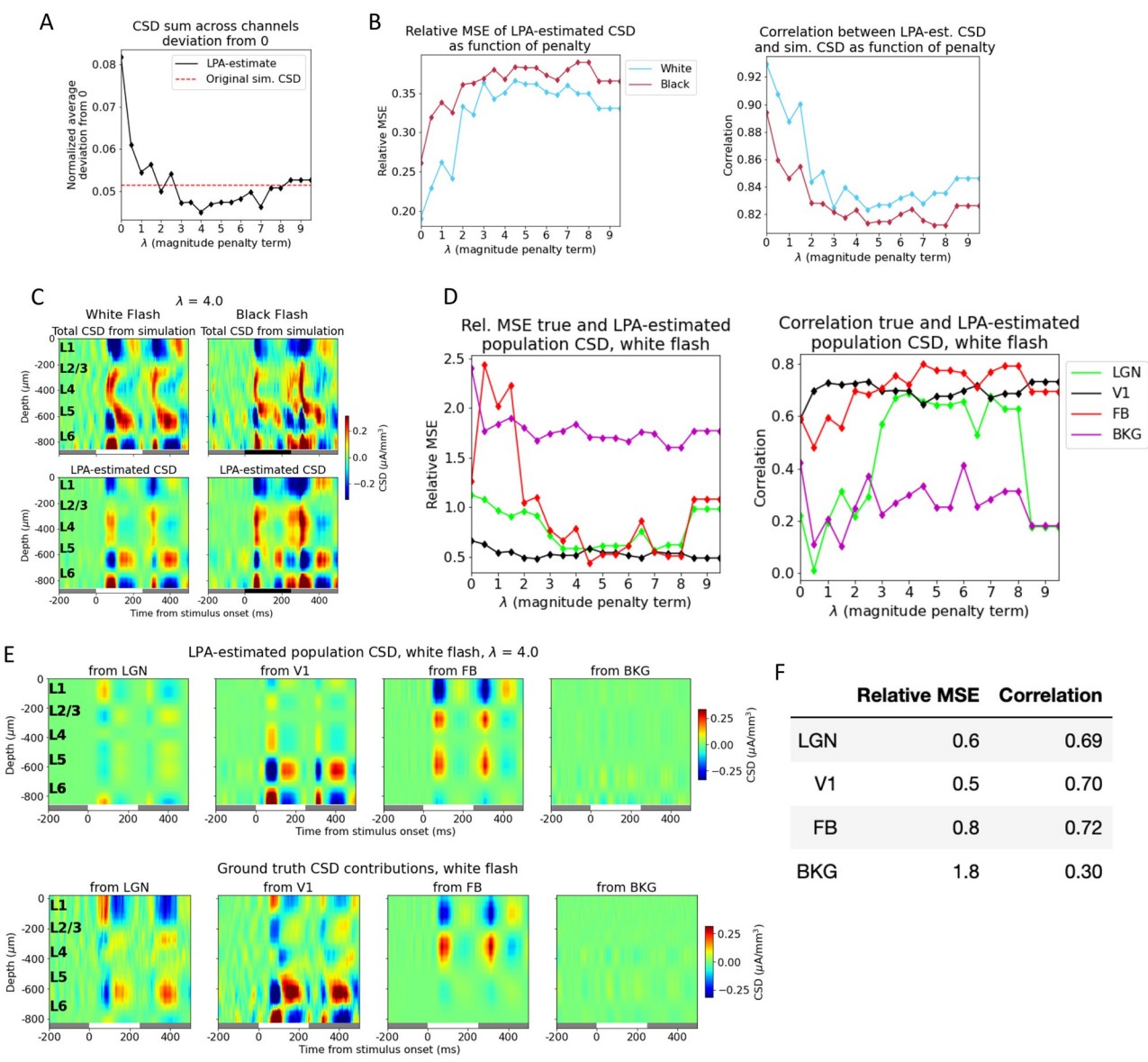

**Fig 5. Penalizing deviations from zero in the summed CSD across channels improves the LPA prediction.** (A) Average sum of LPA-estimated total CSD across channels (black line) with varying penalty (λ) on deviations from zero. Red dashed line: Average deviation from zero of sum across channels for total simulated CSD to which LPA was applied. (B) Relative MSE (left) and correlation (right) between CSD from simulation and CSD estimated from LPA-components at different penalty magnitudes. White flash: blue line, black flash: dark red line. (C) Total simulated CSD (top) and CSD estimated from LPA-components (bottom) in response to white and black full-field flash when the penalty for deviations from zero λ = 4, which is the value at which the deviation from zero for the average sum across channels in the LPA-estimated CSD is smallest (see (A)). White and black bars at the bottom of the plot indicate when the stimulus is presented. (D) Relative MSE (left) and correlation (right) between LPA-estimated and true white flash CSD generated from each presynaptic population at different values of λ. LGN: green line, V1: black line, feedback: red line, background: purple line. (E) Top: LPA-estimated white flash CSD generated from firing in each presynaptic population for λ = 4. Bottom: True white flash CSD generated from firing in presynaptic populations. (F) Relative MSE and correlation between LPA-estimated and true white flash CSD generated from each presynaptic population with λ = 4.

determined to represent specific physiological processes, only that the discovery of these interpretations is less straightforward. We explored whether the CSD of the components obtained from ICA or PCA corresponded well with the true CSD contributions of the four main sources contributing to the simulated CSD (LGN, V1, FB, BKG) by correlating and calculating the

relative MSE between the component CSD and the true CSD contributions. We found a low level of specificity and reliability in the CSD estimated by the components across the flashes.

The second independent component (IC 2) correlated strongly (0.57 and 0.54, respectively) with both the LGN and V1 contribution for the white flash response, but weakly with the CSD from the same structures for the black flash response (0.15 and 0.2, respectively) (S9C and S9G Fig). The third independent component (IC 3) had the strongest correlation with the FB for both the white and black flash responses (0.56 and 0.41, respectively), indicating that this component may be reliably capturing some of the CSD patterns generated by the FB. However, visual inspection of the CSD of this component and comparison to the true CSD from FB reveals that there are several spurious sinks and sources in the deep layers, especially for the white flash. The relative MSE between the IC 3 and true CSD for the white flash response was substantial (3.02) and is likely to be due to these discrepancies.

In the principal components analysis, we used the four components that explained the most of the variance (PC 1–4: 63.9%, 25.6%, 5.1%, 1.6%; cumulative variance = 96.2%) and compared the CSD of these components to the true CSD. The true V1 and FB generated CSD correlates the most with the first principal component (PC 1) CSD for the white flash response (0.42 and 0.55, respectively), while the true LGN CSD correlates the most with the second principal component (PC 2) for the white flash (0.64). For the black flash however, the true CSD correlates the most with the PC 1 CSD for all three structures LGN, V1, and FB (0.29, 0.47, and 0.58), and the PC 2 CSD correlates only weakly (0.12) with the true LGN CSD, even though this component correlated strongly with the true LGN CSD for the white flash. Furthermore, none of the components has a higher correlation than 0.07 with the true BKG CSD. Together, these results suggest that significant proportions of the CSD generated by LGN, V1, and FB CSD are put in the PC 1 CSD, though some of the LGN CSD may be represented by PC 2 for the white flash.

The lack of specificity and consistency across flashes makes it difficult to connect the component CSD to specific structures, and it also suggests that the CSD from the four different main contributors to the simulated CSD is not well separated and identified with neither ICA nor PCA.

## Applying LPA to experimental data

After validating the LPA method on simulated extracellular potentials, we demonstrate its use on experimentally recorded MUA and LFP. The Visual Coding dataset made publicly available by the Allen Institute in 2019 contains electrophysiological recordings across cortical and subcortical areas with a focus on visual structures from 58 mice while they were exposed to a battery of visual stimuli [4]. Nine mice were excluded from the dataset because fading in the fluorescent dye or artifacts in the optical projection tomography (OPT) volume meant that the exact location of the probe penetrating V1 could not be recovered [4] (see Materials and methods). Furthermore, we set a criterion that an individual animal had to have extracellular potentials recorded in LGN, V1, and at least one of the higher visual areas LM, RL, AL, PM, or AM during the full-field flash stimuli presentation to be retained. This criterion was set to ensure a minimal representation of the externally and internally generated activity in the LPA analysis. 27 animals had recordings in LGN, 49 animals had recordings in V1 where exact probe location could be recovered, and of the animals that had recordings both in LGN and V1, 24 animals had recordings in at least one higher visual area. Thus, 24 animals were retained for LPA analysis.

For all animals, we applied LPA with a single population assumed for each structure, as was done for the simulation data in Figs 4 and 5. The reason was that the high synchrony across

laminar populations observed in the simulations for these stimuli could also feature in the experiments, and similarly thwart the ability of LPA to separate the contributions from different layers to the LFP/CSD. The first step of the LPA algorithm (Fig 1E), where the spatial profile of the structure and temporal profile of the firing rate of that structure are obtained from the MUA, was applied to all visual structures for an example animal in the dataset (animal ID: 757970808) in Fig 6A–6G. This animal had recordings of extracellular potentials in five visual areas: LGN, V1, RL, AL, and AM. The recorded MUA in each structure in response to full-field white and black flashes lasting 250 ms is shown in the top panels with the total MUA estimated back from the LPA components below in Fig 6A–6E. Inspecting the resulting temporal profiles for each structure, we observe in Fig 6F a pronounced response to both stimulus onset and offset about 50 ms after stimulus change. Thus, the evoked firing response appears to be captured for each structure in this animal. The relative MSE and correlations between the LPA estimates and the original recorded MUA for each structure are marked with red stars for this animal in the box plots in Fig 6H and 6I. The box plots here show the distributions of relative MSEs and correlations in all visual structures for all 24 animals.

As for the simulated data, the temporal profiles of the firing rates estimated for each structure are convolved with kernels to produce the temporal profiles of the CSD generated by each structure (Fig 7A). We also applied a penalty to the deviations from zero in the CSD summed across channels. Notably, the deviation from zero did not reach the same value as for the recorded CSD for this example animal, even though it did decrease from its value of 0.24 at $\lambda = 0$ (Fig 7B). The relative MSE of the total CSD estimate increased and the correlation decreased when the penalty increased, but not monotonously (Fig 7C). The deviation from zero in the CSD sum across channels was smallest for $\lambda = 1.0$, where it was 0.18. We therefore used the LPA-estimate at this penalty value (Fig 7D) and obtained the spatial profiles of the CSD (Fig 7E). These spatial profiles were multiplied with the temporal profiles to produce estimates of the contribution to the CSD recorded in V1 arising from firing in each structure (Fig 7F). We observe early sinks generated in L2/3 and L4 from LGN, as we did for the simulation (Fig 5), and a deep layer dipole in L5/L6 generated by firing in V1. LPA also predicts dipoles in the upper layers (L1 to L4) generated by recurrent activity in V1, which we only saw with weaker magnitudes relative to the deep layer sinks in the simulation. We moreover observe a strong sink in L1 and L2/3 with a strong source below it in L2/3 and L4 for the contribution from the higher visual area AL, as was observed for the feedback from LM in the model. For RL the predicted contribution is more evenly distributed across the layers, while for AM it is more concentrated in L5/L6. However, the magnitude of the contribution is lower for both these layers than it is for AL.

It may be sensible to sum the contributions from the different higher visual areas together to get a joint estimate for the contribution from feedback. When we do that, we see an upper layer dipole with a sink at the top in L1/L2/3 and a source below it in L2/3 and L4 and a sink below in L5/L6 (Fig 7G). The prediction of a sink generated by feedback in the upper layers, L1 and L2/3, with a source below it in L4 and parts of L5, aligns with the predicted CSD contribution from feedback in the model. The magnitude of the estimated major sink in L5 generated by feedback in the experiment deviates from the magnitude predicted by simulations, but the experimental prediction may reflect synaptic innervation from HVAs in this layer, which is suggested anatomical data [71]. In summary, while there are important discrepancies, the LPA-estimated positions of prominent sinks and sources generated by different populations largely align with their expected distributions based on anatomical data and the predictions made with the model. More detailed data are required to determine the validity of the LPA estimates for experimental data, but the preliminary comparison to model predictions and

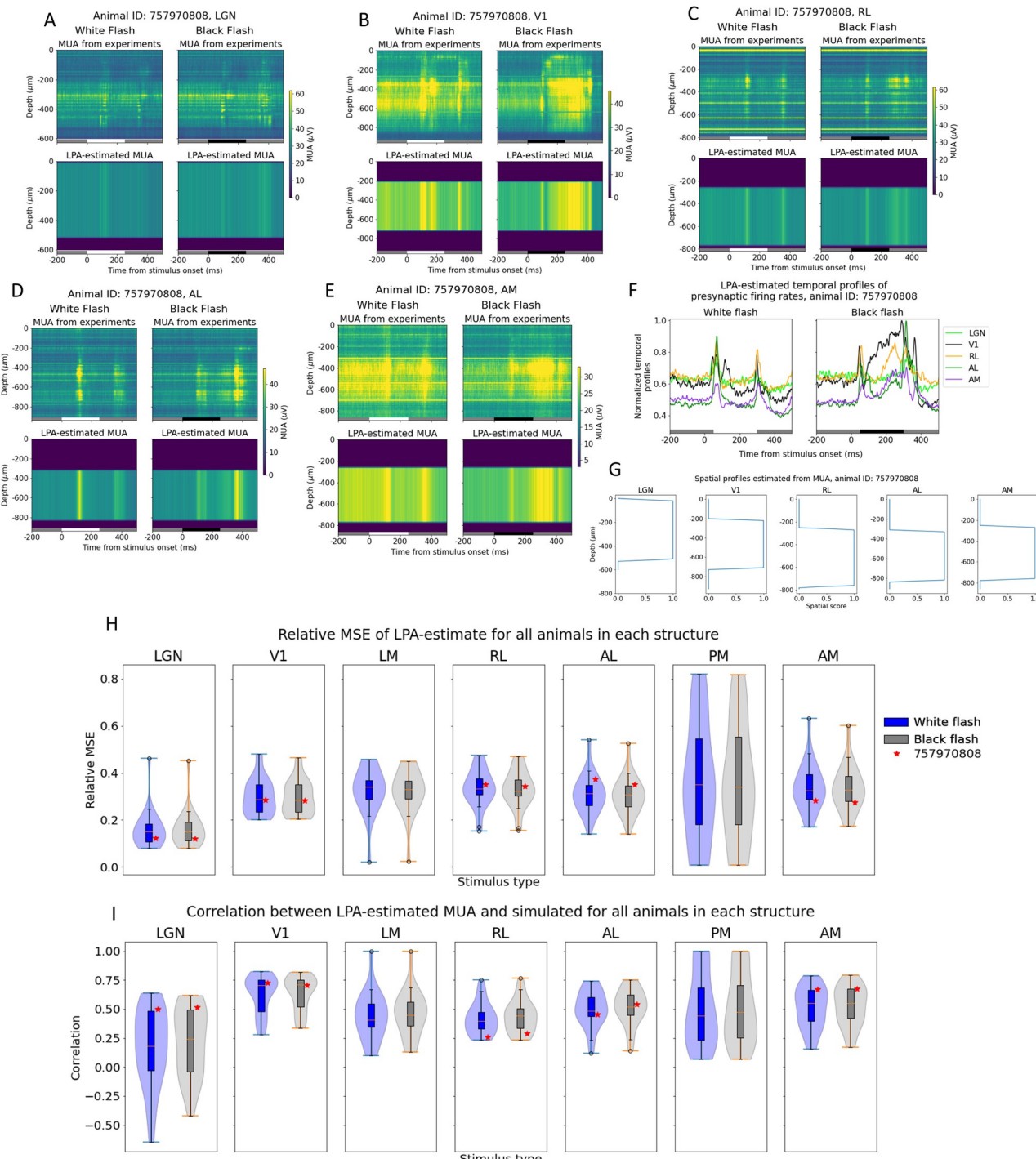

**Fig 6. Applying LPA to MUA from example animal.** (A-E) MUA recorded from visual structures LGN, V1, RL, AL, and AM in an example animal (top) during presentations of white (left) and black (right) full-field flash stimuli and MUA estimated from LPA-components in each structure (bottom). White and black bars at the bottom of the plots indicate when the stimulus is presented. MUA averaged across 75 trials of each flash type. There were no readable recordings from the visual structures AL or AM for this animal. (F) Temporal profile of presynaptic firing rates in each structure estimated with LPA. (G) Spatial profiles of LPA-components along the probe axis in each structure. (H-I) Distribution of relative MSE (H) and correlation (I) between recorded MUA and MUA estimated from LPA-components in each visual structure for all 24 animals. Example animal from panels (A-G) is marked with a red star. Blue box plots and violin plots: white flash, grey box plots and violin plots: black flash. LGN: N = 24, V1: N = 24, LM: N = 20, RL: N = 23, AL: N = 21, PM: N = 18, AM: N = 23.

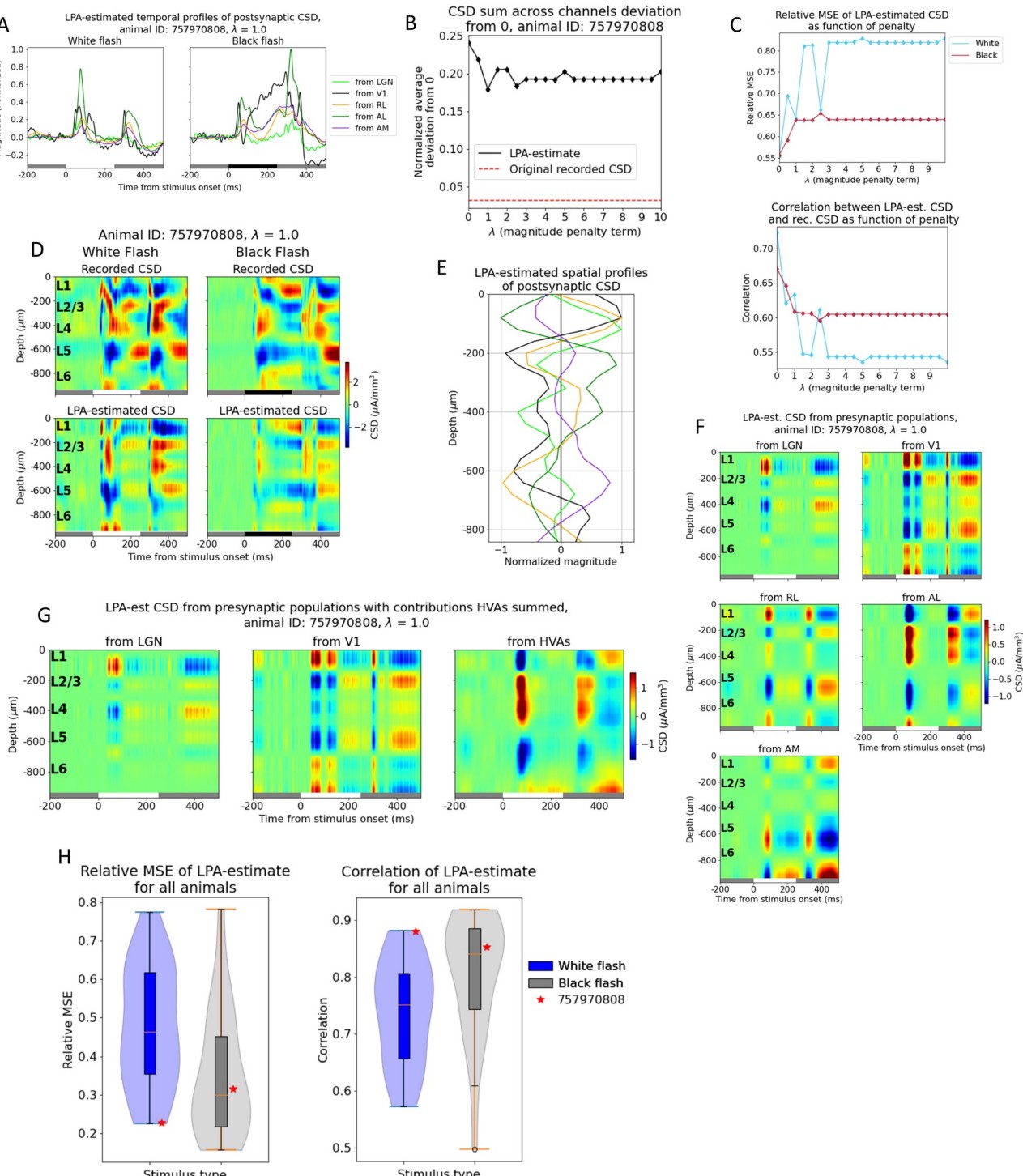

**Fig 7. Applying LPA to CSD from example animal.** (A) Temporal profiles of CSD of each structure in response to white and black flash for an example animal. (B) Deviation from zero for CSD summed across channels for LPA-estimate (black line) and recorded CSD (red dashed line) for an example animal in dataset with varying penalty (λ). (C) Relative MSE (top) and correlation (bottom) between recorded CSD and CSD estimated from LPA-components at different penalty magnitudes. (D) Recorded CSD (top) and CSD estimated from LPA-components in response to white and black flashes when the penalty for deviations from zero λ = 1.0, which is the value at which the deviation from zero is smallest (see (B)). CSD averaged across 75 trials. White and black bars at the bottom of the plots indicate when the stimulus is presented. (E) Spatial profiles of CSD along axis of simulated recording probe generated by each structure. (F) LPA-estimated CSD generated from firing in each presynaptic population for λ = 1.0. (G) LPA-estimated CSD generated from firing in presynaptic populations for λ = 1.0 where the contributions from the different HVAs (RL, AL, and AM) have

been summed. (H) Distribution of relative MSE (left) and correlation (right) between recorded CSD and CSD estimated from LPA-components in V1 for all 24 animals. Example animal from panels (A-F) is marked with a red star. Blue box plots and violin plots: white flash, grey box plots and violin plots: black flash.

anatomical data is at least an indication that the estimated population contributions are reasonable.

## Discussion

In this study, we have utilized spikes and extracellular potentials simulated with a large-scale, biophysically detailed model of mouse V1 to develop and validate the LPA method. We have also demonstrated its use on experimentally recorded extracellular potentials from the mouse visual system. We have found the following:

- The LPA method can be used to identify the positions of layers in a laminar structure and estimate the temporal profile of the layer population firing rates from MUA.

- The LPA method does identify some, but not all contributions from the different laminar populations to the LFP/CSD. This is likely due to excessive high synchrony in firing rates across layers when a fixed inter-stimulus interval protocol is used.

- After introducing a regularization term that penalizes deviations from zero in the CSD summed across channels, the contributions from different brain areas to the CSD are well estimated with LPA.

### Estimating layer positions and firing rates from MUA

We assessed the classification of layer positions by calculating the confusion matrix and the precison, recall, and F1 scores for each layer. The F1 score, which is an aggregate measure of the other classification metrics, was 0.86 or higher for all layers in the model except L4, for which it was 0.67. The somewhat lower score for L4 was caused by the LPA-estimated spatial profile for L4 bleeding over into the neighboring L2/3 and L5, such that some channels that resided in those layers near the border to L4 were incorrectly estimated to be localized in L4 (Fig 2B). Differences in MUA power are the primary determinant of positions of spatial profiles in LPA [62], so the likely cause of the broader LPA-estimated spatial profile for L4 is the high MUA magnitudes close to the peak magnitude in L4 around the borders of L4, as can be observed in S7 Fig. For all other layers, the precision and recall scores were both close to 1.

The organization, cell type composition, thickness, and other properties of layers vary between different cortical structures and between species. For example, in primate V1, layer 4 comprises three different sublayers and makes up a larger proportion of V1 thickness than layer 4 in rodent V1 [72]. The level of differentiation between layers, such as layer 4 vs. layer 5, etc., also varies across species, where layers in primate cortex are more distinct than in rodent cortex. One would therefore expect an even better layer identification, especially of layer 4, from primate data. Thus, an interesting future direction would be to test the application of LPA to ECP data recorded in other species, such as primates.

Altogether, the results from the present application of LPA to rodent data still demonstrate that LPA can be used to estimate the positions of layers in electrophysiological recordings, either as a supplement to histological information or as a substitute when histological data is unavailable.

Multiple studies have been conducted to investigate whether one can correctly identify population firing rates from MUA [43, 73]. Pettersen et al. (2008) [43] demonstrated that the firing rate of a population of L5 excitatory cells scale to the power 3/4 of the MUA and can be accurately estimated from the MUA when that relationship is exploited. Keller et al. (2016) [73], on the other hand, found that when simulating the MUA from medial spinal neurons and fast-spiking interneurons together, the overall accuracy of the population firing rates estimated from MUA did not exceed 50%. However, their goal was to investigate whether the firing rates of two populations could be distinguished and estimated from frequency-differences in their MUA-contribution, which is a somewhat different problem than identifying firing rates from single or non-overlapping populations.

In the present study, we necessarily had to limit our estimates to reproducing the *temporal profiles* of the population firing rates; i.e. the timing of the onset and offset of the evoked response, the peak response, the magnitude of the peak firing rate relative to the baseline, and the magnitude of the firing rate of a given population relative to other populations. The MUA magnitude estimated from multiplying the temporal and spatial profiles of the LPA components is however comparable to the MUA magnitude from the model. Since the goal is not to reproduce the actual amplitudes of the firing rates, our target is easier to reach, and that is probably why we achieved relatively good correspondence between the temporal profiles of the LPA-components and the underlying laminar population firing rates of the model for the different layers (Fig 2E). The estimation of temporal profiles and positions of laminar populations from MUA is a beneficial application of LPA independent of whether users intend to also analyze LFP.

## Estimating laminar contributions to the LFP/CSD

The contributions from each laminar population could not be satisfactorily separated and identified with LPA, even though the contributions from some populations were partially recapitulated (Fig 3D–3F). This inability to identify the contributions from some of the layers is likely caused by the high synchrony in firing rates across these populations (Fig 3G). The temporal profile of the firing rates is the only distinguishing feature between the different presynaptic populations in the decomposition of LFP/CSD in LPA, so if the correlation between the firing rates of different populations is high, one can expect cross-contamination in the CSD attributed to the different LPA-components. We experimented with different numbers of populations assumed between 1 and 5 for V1 (S11 Fig), but the best results were achieved when all the laminar populations of V1 were merged together into a single population.

The high synchrony observed here might in part be diminished by the use of other stimulus protocols. Protocols with randomized and variable inter-stimulus intervals have been demonstrated to increase statistical efficiency in estimating the event-related hemodynamic response in functional magnetic resonance imaging experiments [74]. A similar stimulus protocol in electrophysiological experiments could improve the estimates of the LFP/CSD too, and increase our ability to disentangle contributions from different parts of the visual (or other) system. In this study, we are constrained by the protocol used in the experiments since we are using experimentally recorded spike trains as input for both LGN and LM. The spike trains were recorded during presentations of full-field flash stimuli lasting 250 ms and with fixed intervals between each presentation. The duration of the stimulus limits the frequency with which stimuli can be presented. The maximal frequency of stimulus presentations would have to be lower than 4 Hz, which may be too low to cover the temporal scales of stimulus presentation where the responses of different populations can be untwined. A future extension of this

study would be to carry out experiments with a stimulus protocol that allows for optimal system identification both in experiments and the model.

## Estimating contributions from different structures to the LFP/CSD

Based on the observation (and expectation) that the synchrony between different brain areas is lower than the synchrony between layers within the same structure, we investigated whether the CSD generated by the different brain areas could be reliably distinguished with LPA. We found that the estimated contribution from V1 as a whole was signifcantly better than the estimated contributions of the individual laminar populations (Fig 4E and 4F). The correlation was substantially higher and the relative MSE substantially lower between the estimated and true V1-generated CSD than between the estimated and true CSD generated by the individual layers (Fig 3D–3F).

However, even though the salient sinks and sources observed in the true CSD from LGN and feedback were recapitulated in the LPA-estimated CSD from these structures, there were also sinks and sources in the estimate that were not observed in the true CSD. Additionally, the relative magnitude of different sinks and sources in the CSD pattern did not always match the magnitudes in the true CSD. We speculated that this might in part be alleviated if we penalized imbalances in the CSD across channels for the LPA estimates, and therefore expanded the cost function for the CSD to include a penalty on deviations from zero in the CSD summed across channels for the LPA-estimated CSD of each structure (Eq 10 in Materials and methods).

We discovered that the correspondence between the LPA-estimated and true population CSD was optimal at the values of λ (the penalty control parameter) for which the deviation from zero was smallest (Fig 5A and 5D–5F). At λ = 4.0, the value where the deviation from zero in the CSD sum was minimal, the CSD estimates improved for all populations on both measures with the singular exception of the correlation for the background. The spurious sinks and sources in the estimates are not eliminated completely and there are still deviations in magnitudes for the sinks and sources that have the right position and timing, but these discrepancies are diminished and the most salient features in the true CSD patterns are recapitulated with the introduction of the penalty. Moreover, the abovementioned amendment to the stimulus protocol could help improve the LPA estimates further for the whole structures as well as for the individual layers of V1.

To compare our results from the decomposition with LPA with other decomposition methods, we tested the application of ICA and PCA on the simulated data utilized here (S9 and S10 Figs). We found that neither method was able to reliably disentangle and reproduce the CSD generated from the four main contributors (LGN, V1, FB, BKG).

It was pointed out in [19] and [58] that the underlying statistical assumptions of ICA and PCA may not be valid for LFP data in all structures or experimental conditions. While PCA assumes the different generators of LFP to be orthogonal, ICA assumes them to be statistically independent. In the case of mouse V1, LGN provides input to all layers of V1, which means that there can be significant contributions from LGN to several of the sinks and sources recorded in V1. Furthermore, the contributions that arise from recurrent activity within V1 depend on prior input from LGN, so the V1 contribution cannot be said to be generated independently from LGN. The same applies to the contribution from higher visual areas, which depends on prior input from V1. Thus, for a structure like V1, the assumption of statistical independence is unlikely to be valid. (ICA was used to decompose LFP and identify layers in mouse V1 in [15]. However, it should be noted that the decomposition was limited to LFP in the gamma range in this study. The assumption of statistical independence may be more

applicable in restricted frequency ranges.) Additionally, the components obtained with PCA or ICA are not straightforward to interpret in terms of the underlying physiology.

The LPA-estimated population CSD achieved higher correlations and lower relative MSE than PCA and ICA. Moreover, the problems with interpretability and difficulty in identifying which component corresponds to which population are avoided altogether since the identity and interpretation of each component is built into the LPA method and can be obtained directly. It also circumvents potential violations of statistical assumptions like orthogonality or statistical independence of the different generators of LFP, and can be used on LFP data from structures where such assumptions are not valid.

It should be noted that even though the LPA method has an advantage over ICA and PCA in that data from external recordings can easily be included in the decomposition, it is not given that LPA must necessarily perform better in all conditions and structures. For example, there may be neural data for which different sources of LFP generation are truly statistically independent. In these cases, ICA can disentangle LFP contributions as well as LPA, even if recordings from external structures are not utilized in the decomposition with ICA.

## Applying LPA to experimental CSD

We applied LPA with the same conditions as in Fig 5 on V1-recorded LFP from the experimental dataset, i.e., with only a single population for the whole of V1 and penalizing deviations from zero in the CSD summed across channels for the population estimates, since that was what produced the optimal results in the model. In the example animal shown in Fig 6, we observed that the temporal profiles of the firing rates in each structure estimated with LPA exhibited prominent and largely transient evoked responses about 50 ms after flash onset and offset, as was observed in the model. The black flash response in V1 was an exception here, as the response to flash onset was more sustained for this structure and stimulus condition. However, we can observe in Fig 6B that the MUA amplitude after flash onset is also sustained rather than transient. Thus, the LPA-estimated sustained temporal profile is likely to simply reflect the recorded MUA here. On the whole, these results indicate that the temporal profiles of the firing rates in each structure can be estimated using the LPA method in experiments as well.

A notable result in the box plot of correlations between recorded and LPA-estimated MUA for LGN is that for some (8) animals, the correlation is negative (Fig 6H). In other words, the LPA-estimate does not capture the MUA well for these animals. We investigated whether these animals were different from other animals in the dataset in other ways, and found that they all had relatively few recording channels in LGN compared to most other animals with extracellular potentials recorded in LGN. In S8 Fig, the number of recording channels in LGN for each animal is plotted against the correlation between the recorded and LPA-estimated MUA. In this plot, we can observe that the animals with negative correlations typically have fewest recording channels in LGN. We conducted a linear regression analysis to quantitatively evaluate the relationship between number of recording channels in LGN and the fit of the LPA-estimated MUA, and found that the correlation was $r = 0.77; p = 9E − 6$. This suggests that the poor fit of the LPA-estimate for these animals is in large part due to limited information about the MUA in LGN stemming from a low number of recording channels in the structure.

Applied to the experimental CSD, we observed a slight reduction in the deviation from zero in the summed CSD across channels for the LPA-estimated CSD (Fig 7B). This was typical for the application of penalty to the deviations from zero in the experiments (S12 Fig). This is markedly different from the effect of applying LPA with penalties on deviations from zero to the model CSD, where the reduction in the deviation from zero with increasing penalty was

more pronounced (Fig 5A). The weaker effect on the LPA-estimated experimental CSD could be due to the more limited information about the input to V1. We do not have recordings from all the structures that provide input to V1 in the experiments, which limits the precision with which the contribution from each structure can be estimated. A CSD contribution that should be attributed to a structure that was not recorded (during the presentation of flash stimuli) may be attributed to one of the other structures, which will make the estimates of CSD generated by each structure more inaccurate. And this, in turn, can potentially exacerbate imbalances in the sinks and sources across channels for the population estimates and make it harder to reduce deviations by penalizing it. In the model, on the other hand, we have full knowledge of all inputs to V1 and avoid this particular issue.

Nonetheless, some interesting observations can be made from the LPA-estimated population CSD from the experimental recordings (Fig 7G). If we sum the estimated contributions from all higher visual areas (RL, AL, AM) for which we have recorded MUA in the example animal, we observe that the estimated contribution from the higher visual areas has a sink in the supragranular layers L1 and L2/3 and a source below it in L2/3 and L4 after flash onset and offset, as observed for the model (Fig 5E, bottom). We also see a sink in the deeper layers—L5 and the upper part of L6—which we do not have in the model. However, synapses from the higher visual area LM, for example, are known to terminate in L5 in addition to apical tufts in L1 and L2/3 [66, 70, 71, 75, 76], so this deep layer sink estimated from the experimental CSD may reflect those connections. Furthermore, the L4 sink arising after flash onset from LGN input in the model is also present in the estimated contribution from LGN in this example animal. There are bigger differences between the estimated V1 contribution for this animal and the model V1 estimate. In particular, there is a more even distribution of magnitudes of sinks and sources across the depth in the experiment, while in the model there is a greater dominance of the dipoles in L5/L6. However, the deep layer dipole observed in the true CSD generated by recurrent connections in V1 in the model is also present in this example experiment.

## Limitations

Certain limitations with the use of LPA should be noted. First, when the temporal profiles of firing rates are estimated from the MUA, the firing rates of excitatory and inhibitory cells are merged into the firing rates of one population (unless they happen to be spatially separated). Input from excitatory and inhibitory cells will generate currents with opposite signs in the postsynaptic cells; excitatory input will generate a current sink, while inhibitory input will generate a current source [6]. These opposite effects will not be captured by the estimated contribution from a population where the temporal profile reflects spikes from both excitatory and inhibitory cells. In the model, the majority of the MUA was generated by excitatory cells and the primary sinks and sources was previously shown to reflect excitatory input [53]. Therefore, the error from this simplification is expected to be minor for the model. However, this cannot always be guaranteed for other models or for experiments.

As discussed above, estimating only temporal profiles and relative magnitudes of firing rates from MUA is an easier target than estimating both temporal profiles *and* magnitudes of firing rates. However, only estimating temporal profiles is also not trivial, and there will also be inaccuracies in the estimated temporal profiles of firing rates from MUA. The correlation between the LPA-estimated and true temporal profiles of population firing rates was not perfect even though it was high (Fig 2D–2E). These issues will affect the accuracy of the LFP/CSD contributions estimated from the temporal profiles in the next steps too. For the model, we used the firing rates computed from known spikes from all excitatory cells in the decomposition of the CSD. This is not possible in experiments, which means that the limited accuracy in

temporal profile estimates from MUA will constitute a limitation when LPA is applied to experimental data.

The inclusion of input from external structures in the application of LPA of course necessitates that the activity in those structures has been recorded simultaneously with the structure that LPA is applied to. In the application to simulated data, we had full knowledge of the activity in all structures providing input to V1. In the application to experimental data, however, there may be structures that provide a non-negligible input that are not included in the analysis due to a lack of recordings in those areas. This will be a source of error as the contributions from those structures will falsely be attributed to the input from the structures for which we happened to have recordings.

In some experimental conditions, such as when the animal is anesthetized, the effects of external inputs in LFP generation may be more limited [64–66]. In those conditions, the LPA method can have utility even if there are no recordings from external structures. Indeed, in the original presentation of the LPA method, it was applied to experimental data from rat barrel cortex where there was no MUA data for other areas [62]. Those recordings were made while the rat was lightly anesthetized and the timing of the onset of evoked LFP responses and the MUA in the barrel cortex suggested that, in these experiments, most of the LFP was generated by recurrent activity within the barrel cortex. Additionally, the first part of the LPA algorithm, where laminar positions are identified from the MUA, can be applied even if there are no recorded data in external structures and there are no experimental conditions that limit their influence on the structure in question. The identification of laminar positions from MUA data only depends on differing MUA magnitudes across layers, and this application of the LPA method therefore has utility even in experiments with limited simultaneous recordings from different structures. If there are no conditions that can be expected to limit the effects of input from external structures, however, then a lack of simultaneous recordings from those structures would constitute a limitation in which experimental LFP data LPA can be applied to.

Lastly, since the spiking of all neurons in the column can in principle contribute to the measured LFP through their afferent connections, LPA requires an estimate for the spiking for all neurons in the column. The present application of the LPA method was thus limited to full-field flashes, that is, a spatially unpatterned stimuli. In this situation the spiking activity recorded from neurons near the probe in a particular layer can reasonably be assumed to be representative of the spiking activity of all neurons in this layer in the column. With spatially patterned stimuli, even as simple as a grating, this is unlikely to hold true because neurons close to the probe and those laterally further way will in general have different firing activities. In a multi-shank arrangement where spiking activity is recorded at multiple different locations in the column, patterned stimuli may, however, also be appropriate for the application of the LPA method.

## Conclusion

In this study, we have found that LPA can be used to uncover laminar positions and to estimate the temporal profile of laminar population firing rates from MUA. LPA is also able to separate and identify the salient current sinks and sources generated by external inputs—feedforward from LGN, feedback from LM—as well as recurrent connections within V1. The decomposition of LFP/CSD with LPA provides a more accurate representation as well as a more direct interpretation of the components in terms of the underlying circuit mechanisms than traditional, statistical decomposition tools such as PCA and ICA.

## Materials and methods

### Laminar population analysis

The LPA method utilizes MUA and LFP data jointly to spatiotemporally decompose the recorded LFP. The method can be divided into four steps (Fig 1E).

**Decomposing MUA.** In the first step, the recorded MUA is decomposed into $N$ populations (the number is assumed a priori) with spatial profiles displaying the position of each population and temporal profiles showing the time course of the firing rate of each population. Applied to a laminar structure, this decomposition can uncover the position of the different layers and the temporal profile of the laminar firing rates. The MUA estimated from these spatial and temporal profiles can be expressed as:

$$\phi_{M,\text{est}}(z_i, t_j) = \sum_{n=1}^{N_{\text{pop}}} M_n(z_i) r_n(t_j) \tag{1}$$

The form of the spatial profiles is assumed to be non-overlapping trapezoids [62]. The parameters determining their distribution for a population $i$ are their position $z_i$, the width of the top $a_i$, and the width of their slope $b_i$, and the height is set to 1 for all populations. Assuming non-overlapping spatial profiles forces the populations to be localized, in contrast to other spatiotemporal decompostion methods, such as PCA [62], where the positions of the predicted populations can both be spatially discontinuous and overlapping. These assumptions are made because the laminar populations to be uncovered are expected to be continuous and non-overlapping.

The parameters of the spatial profiles are first initialized at random values. Then the temporal profiles of the firing rates are estimated by utilizing the initial guess on the spatial profiles together with a pseudoinverse on the recorded MUA, which can be expressed as follows:

$$\mathbf{r}_{\text{est}} = \left(\mathbf{M}^\dagger \phi_M\right)^{\text{T}} \tag{2}$$

where $\mathbf{M}^\dagger$ denotes the pseudoinverse of an $N_{\text{pop}} \times N_{\text{chan}}$ matrix containing the population spatial profiles of all $N$ populations, $\phi_M$ is the recorded MUA, and $\mathbf{r}_{\text{est}}$ is an $N_{\text{pop}} \times B$ matrix of firing rates over a time period discretized into $B$ time bins for all populations, where each row corresponds to the firing rate of a single population. Once the firing rate temporal profiles have been calculated, an estimate for the MUA given the initial guess on the spatial profiles can be obtained according to Eq 1.

The discrepancy between this estimate $\phi_{M,\text{est}}$ and the recorded MUA $\phi_M$ is evaluated by calculating the relative mean square error $e_M$:

$$e_M = \frac{\sum_{i=1}^{N_{ch}} \sum_{j=1}^{B} \left(\phi_M(z_i, t_j) - \phi_{M,\text{est}}(z_i, t_j)\right)^2}{\sum_{i=1}^{N_{ch}} \sum_{j=1}^{B} \left(\phi_M(z_i, t_j)\right)^2} \tag{3}$$

The parameters $z_i$, $a_i$, $b_i$ defining the spatial profiles of the populations are then optimized until the discrepancy between the MUA estimated from the LPA-components and the recorded MUA evaluated with Eq (1) is minimized.

**Decomposing LFP.** The optimal temporal profiles of each presynaptic population found from the MUA are convolved with a suitable kernel to estimate the time course of the postsynaptic LFP resulting from firing in the presynaptic populations. This is expressed as follows:

$$\mathbf{R}_{\text{est}} = \mathbf{h} \circledast \mathbf{r}_{\text{est}} \tag{4}$$

where $\mathbf{R}_{\text{est}}$ is the time course, or temporal profile, of the postsynaptic LFP and $\mathbf{h}$ is a matrix

with dimensions $N_{\text{pop}} \times B$ with $N_{\text{pop}}$ kernels. Here, as in the original publication describing the LPA method [62], we used exponential functions for the kernels:

$$h_i(t_j) = \frac{1}{\tau_i} e^{-(t_j - \Delta_i)/\tau_i} \Theta(t_j - \Delta_i) \tag{5}$$

where the parameters $\tau_i$ and $\Delta_i$ are the time constants and the delays, respectively, for a population $i$ and $\Theta(t_j - \Delta_i)$ is the heaviside function, imposed by causality: $h_i(t_j - \Delta_i < 0) = 0$. In the original publication of LPA, a single kernel was used for all populations. Here, we allowed for unique kernel parameters $\Delta$ and $\tau$ for each population. This choice was based on the fact that one cannot expect the time course of the effect of firing in each presynaptic population to be the same for all postsynaptic populations, as it will depend on the synaptic properties, synaptic distribution, and the geometry of the postsynaptic cells.

The time courses of the postsynaptic LFP are then utilized together with the recorded LFP to find the spatial profiles $\mathbf{L}_{\text{est}}$ of the population LFP:

$$\mathbf{L}_{\text{est}} = \phi_L \mathbf{R}_{\text{est}}^\dagger \tag{6}$$

where $\phi_L$ is the recorded LFP. Finally, the temporal and spatial profiles of the population LFP are multiplied to estimate the LFP $\phi_{L,est}$ from the components according to:

$$\phi_{L,\text{est}} = \mathbf{L}_{\text{est}}\mathbf{R}_{\text{est}} = \mathbf{L}_{\text{est}}(\mathbf{h} \circledast \mathbf{r}_{\text{est}}) \tag{7}$$

and this estimate is compared to the recorded LFP. The parameters of the population kernels $\boldsymbol{h}$ are optimized (with a differential evolution algorithm in this paper) until the discrepancy between the LPA-estimated and the recorded LFP is minimized. As for the MUA, the cost function for the LFP is the relative mean square error of the estimate:

$$e_L = \frac{\sum_{i=1}^{N_{ch}} \sum_{j=1}^{B} \left(\phi_L(z_i, t_j) - \phi_{L,\text{est}}(z_i, t_j)\right)^2}{\sum_{i=1}^{N_{ch}} \sum_{j=1}^{B} \left(\phi_L(z_i, t_j)\right)^2} \tag{8}$$

In this project, we calculated the CSD from the recorded LFP and applied LPA to that instead. This was done because the CSD is more localized and easier to interpret in terms of the underlying transmembrane currents. The steps to the LPA algorithm are unchanged by using the CSD rather than the LFP.

**Penalizing deviations from zero.**   According to Kirchoff's current law, the sum of currents entering or leaving the extracellular space from a neuron has to be equal to zero. In the traditional method for calculating the CSD from LFP, where the CSD is obtained by taking the double spatial derivative of the LFP and multiplying with -1 and the conductivity, this condition is enforced by design such that the sum of sinks and sources across channels on the recording probe is equal to zero [36]. However, the traditional method assumes constant CSD in the (infinite) plane orthogonal to the probe, which is generally not a valid assumption. Other methods have been developed to calculate the CSD from the LFP that do not rely on the assumption of constant CSD in the infinite plane. Here, we utilized the delta iCSD method, where the CSD is only assumed to be constant within a limited radius from the recording probe. For the model, we assumed constant CSD in discs of radius of 400μm, equivalent to the spatial extent of the biophysical core of the model. For the experiments, we assumed constant CSD within a radius of 800μm, approximately equal to the extent of mouse V1.

The alternative methods for calculating the CSD, such as delta iCSD, do not enforce that the sum of sinks and sources across channels of the recording probe is equal to zero. Therefore, the cost function of the LPA method was here amended to include a regularization term that

penalizes deviations from zero in the sum of CSD across channels. This penalty term is expressed as:

$$penalty = \lambda \frac{1}{B} \sum_{j}^{B} \sum_{i}^{N_{ch}} C_{\text{est}}(z_i, t_j) \tag{9}$$

where $\lambda$ is a control parameter that determines the magnitude of the penalty. The total cost function for the CSD estimate then becomes:

$$e_C = \frac{\sum_{i=1}^{N_{ch}} \sum_{j=1}^{B} \left(C(z_i, t_j) - C_{\text{est}}(z_i, t_j)\right)^2}{\sum_{i=1}^{N_{ch}} \sum_{j=1}^{B} \left(C(z_i, t_j)\right)^2} + \lambda \frac{1}{B} \sum_{j}^{B} \sum_{i}^{N_{ch}} C_{\text{est}}(z_i, t_j) \tag{10}$$

## V1 model

There are in total 230,924 neurons in the model. 51,978 of these are biophysically detailed multicompartment neurons and they form a cylindrical core with diameter of 800μm and height 860μm. The remaining 178,946 neurons are leaky-integrate-and-fire (LIF) neuron models which form an annulus with thickness 445μm and the same height surrounding the core of biophysical neurons [63]. In layers 2/3 to 6, there is one excitatory class and three inhibitory classes (Pvalb, Sst, Htr3a) unique to each layer. In layer 1, there is only one Htr3a inhibitory class and no excitatory neurons.

There are three sources of input to the model representing LGN, LM, and the background input from the rest of the brain. The LGN module is composed of 17,400 units that are connected to the excitatory classes and the Pvalb classes in all layers and the Htr3a class in L1. The LM source is a single node that provides input to the excitatory, Pvalb, and Sst classes in L2/3 and L5 and represents the feedback from higher visual areas. The background is also a single node, and it is a Poisson source firing at 1 kHz providing input to every neuron in the V1 model and represents the influence of the rest of the brain.

**Dendritic targeting.**   LGN to V1

Synapses from LGN onto inhibitory V1 neurons are placed on the soma and anywhere on the basal dendrites [53, 63]. Synapses targeting excitatory V1 neurons are placed within 150μm from the soma on apical and basal dendrites.

V1-V1

Synapses for recurrent connections are positioned as follows:

*Excitatory-to-Excitatory Connections*

Excitatory to excitatory synapses are placed on the basal and apical dendrites and avoid the soma. They are placed anywhere along the dendrites in layers 2/3 and 4, while in layers 5 and 6 they have to be within 200μm and 150μm from the soma, respectively.

*Excitatory-to-Inhibitory Connections*

Excitatory to inhibitory synapses are placed on the somata and the basal dendrites with no distance limitations.

*Inhibitory-to-Excitatory Connections*

Pvalb to excitatory synapses are placed within 50μm from the soma on both basal and apical dendrites. Sst to excitatory synapses are placed exclusively more than 50μm from the soma. Htr3a to excitatory synapses are placed between 50μm and 300μm from the soma.

*Inhibitory-to-Inhibitory Connections*

Inhibitory to inhibitory synapses follow the same rules as inhibitory to excitatory synapses described above.

<u>LM-V1</u>

The LM to V1 synapses connect to the region within 150μm from the soma of the apical dendrites of L2/3 excitatory neurons and the basal dendrites of the L5 excitatory neurons as well as the the apical tufts (> 300μm from the soma) of L5 excitatory neurons. They also connect to the Pvalb and Sst inhibitory cells in the same layers at the soma and on the basal dendrites at any distance from the somata.

## Stimulus protocol

The stimulus used both in the experiments and the model was fullfield white and black flashes. White or black screens are presented for 250 ms with a gray screen between stimulus presentations. In the experiments, the interval between flash presentations is 1.75 s and each flash type is presented for 75 trials. In the simulations, the interval between flash presentations is 0.5 s, and each flash type is presented for 10 trials. The input from LGN and LM in the simulations was constructed from spike trains recorded experimentally in these structures during presentations of the abovementioned stimuli [53].

## Simulation

We utilized data from simulations on the final model presented in [53], and the output data and the files and code necessary to simulate this model version are provided in the Materials and Methods section of that paper. The model was built and the simulations carried out with the Brain Modeling Tool Kit (BMTK) software [77]. Instructions on how to run simulations of the model are provided in [63].

## Ground truth data

The CSD contributions from the various populations and structures that served as the ground truth for testing and validating the LPA decomposition were obtained from a simulation where the connections from all populations except the one of interest removed. For example, to get the contribution from LGN to the CSD, all recurrent connections within V1 were cut and the input from LM and BKG removed, such that all postsynaptic LFP in V1 during the simulation could only stem from the geniculate input. The same approach was followed to get the contributions from LM and BKG.

Obtaining the contribution from recurrent activity within V1 is less straightforward, since all external input is removed to eliminate its contribution to the LFP/CSD, and without the external input there is nothing that can initiate the recurrent activity in V1 that we want to extract the contribution from. However, a feature in the BMTK, allows the user to run a simulation where they replay stored spikes. After a simulation on the full V1 model with all inputs from external populations intact, the spikes of all V1 neurons are stored to file. Then, a simulation can be run where the stored spikes of all V1 neurons are replayed even in the absence of any external input. Thus, with all the external input removed, the resulting LFP/CSD could only have been generated by the replayed spikes of the V1 neurons, and the contribution from recurrent activity within V1 has been isolated. To obtain the contribution from an individual layer (or cell class), one can either only replay the spikes from this population or cut all connections from other populations. The feature is described in more detail here: https://alleninstitute.github.io/bmtk/tutorial_bionet_disconnected_sims.html.

## Data processing

The spike and extracellular potential data from simulations on the V1 model was obtained from the publicly available dataset in [53]. The experimental spike and extracellular potential data was obtained from the publicly available Visual Coding dataset [4].

**MUA, LFP, and CSD.**   To get the LFP, the extracellular potential was low-pass filtered with a cutoff frequency of 300 Hz using a 5th order Butterworth filter and the functions scipy. signal.butter and scipy.signal.filtfilt in Python. The MUA was obtained by utilizing the high-pass version of the same 5th order Butterworth filter with a cutoff frequency of 300 Hz. Other, higher values for the cutoff frequency for the MUA have been used previously (e. g. [43, 62]). However, it has been shown that spikes can contribute significantly to frequencies as low as 100 Hz [73]. On the other hand, lower cutoff frequencies will mean that there will be a greater part of the MUA signal will come from biophysical processes associated with LFP. Thus, the cutoff frequency of 300 Hz was here chosen to strike a balance between losing spike information and including some LFP frequencies in the MUA. Other cutoff frequencies were tested, but only produced minor changes in the results.

The CSD was calculated from the LFP using the delta iCSD method [36] for both the simulated and the experimentally recorded data. With the delta iCSD method, the CSD is assumed to be laterally constant within a certain radius from the recording probe. Applied to the simulation data, the radius was set to 400µm, equal to the extent of the core region of the V1 model where the biophysically detailed neuron models reside. Applied to the experimental data, the radius was set to 800µm, approximately equal to the size of mouse V1.

**Firing rates.**   The population firing rates were obtained from the spike data by averaging the spike count over all cells in each population with a bin size of 1 ms and filtered with the function scipy.ndimage.gaussian_filter with sigma = 2. They were then averaged over 10 trials of each flash type in the simulation data and 75 trials of each flash type in the experimental data.

## Experimental data

### Quality control

Due to fading of the fluorescent dye or artifacts in the OPT volume, the exact probe location in V1 could not be recovered for 9 out the 58 animals in the Visual Coding dataset [4]. For an animal to be retained for the LPA analysis, we set a criterion where it had to have extracellular potentials recorded in LGN, V1, and at least one of the higher visual areas LM, RL, AL, PM, or AM during presentation of the fullfield flash stimuli. The reasoning behind setting this criterion was that it would ensure at least a minimal representation of primary generators of LFP in V1. 27 out of the 58 animals had recordings in LGN, 49 had recordings in V1, and out of the animals that had recordings in both LGN and V1, 24 had recordings in at least one higher visual area. This meant that 24 animals were retained for LPA analysis.

### Quantitative analysis

**Confusion matrix, precision, recall, and F1 score.**   The confusion matrix and the associated precision, recall, and F1 scores are used to assess classification algorithms [67–69]. In a confusion matrix, the true categories of observables lie along one axis, while the predicted or estimated categories lie along the other axis. The number of estimated classifications of each observable are tallied in the appropriate element of the confusion matrix. In the context of estimating the laminar position of electrodes, with estimated layer position on the y-axis and true

layer position on the x-axis (as in Fig 2C), this means that +1 is added for each electrode to the element on the row corresponding to the layer the electrode is predicted to belong to and the column the the electrode actually belongs to. For example, if an electrode is predicted to belong to L2/3, but the true layer it resides in is L4, +1 would be added to the second row from the top, corresponding to the L2/3 predicted row, and the third column, corresponding to the true L4 column. Thus, the estimated and true laminar positions coincide along the diagonal. Finally, the elements along each row can be normalized by the sum of the elements in that row to get the proportion of correct and incorrect classifications for each layer (category).

The metrics precision and recall are calculated from the confusion matrix. Precision is defined as the ratio of true positives (TP) to the sum of true positives and false positives (FP)—the total number of positives:

$$precision = \frac{TP}{TP + FP} \tag{11}$$

The precision score is calculated for each layer separately, such that one gets $N$ precision scores where $N$ is the number of populations. The true positives are the elements along the diagonal of the classification matrix. The false positives are the off-diagonal elements in the row of the laminar population the precision is calculated for.

Recall is defined as the ratio of true positives to the sum of true positives and false negatives (FN):

$$recall = \frac{TP}{TP + FN} \tag{12}$$

The recall score is also calculated for each layer separately, and the false negatives will be the off-diagonal elements in the column of the true layer. For example, if the recall is calculated for L4, the false negatives will be the elements in the L4 column that are not estimated to be in L4 by LPA.

The precision gives a measure of the extent of type 1 errors in the classification, while the recall gives a measure the extent of type 2 errors. The F1 score gives a balanced metric of the precision and recall scores, and is defined as the harmonic mean of the two:

$$F_1 = \frac{2 \times precision \times recall}{precision + recall} \tag{13}$$

**Correlation.** Correlation was used to assess the similarity between the temporal profiles of the population firing rates estimated with LPA and the true temporal profiles of the population firing rates of the model. It was also used to compare the CSD patterns estimated with LPA and the true CSD patterns of the model, both for the whole of V1 and for the individual populations. In both cases the Pearson correlation coefficient was calculated using the function numpy.corrcoef.

## Supporting information

**S1 Fig. MUA variance by cell types.** (A) MUA contribution of cell type families in the model. (B) Proportion of variance explained by excitatory (left) and inhibitory (right) cell type families.
(TIF)

**S2 Fig. Black flash estimates from LPA on simulated CSD with V1 laminar populations distinguished.** (A) Top: Black flash CSD generated from firing in each presynaptic population

estimated with LPA using temporal profiles obtained from MUA as shown in Fig 2D. Bottom: Relative MSE and correlation between LPA-estimated and true black flash CSD contributions from each presynaptic population. (B) True Black flash CSD contributions from firing in each presynaptic population. (C) Top: Black flash CSD generated from firing in each presynaptic population estimated with LPA using temporal profiles obtained from firing rates of excitatory cells in the model in each layer from 2/3 to 6 and inhibitory cells in layer 1. Bottom: Relative MSE and correlation between LPA-estimated and true black flash CSD contributions from each presynaptic population.
(TIF)

**S3 Fig. Black flash estimates from LPA on simulated CSD with one population for V1.** (A) Top: LPA-estimated black flash CSD generated from firing in each presynaptic population. Firing rates of excitatory cells in the model are used for the presynaptic temporal profiles of V1 in the decomposition. (B) True black flash CSD generated from firing in the same presynaptic populations. (C) Relative MSE and correlation between LPA-estimated and true black flash CSD generated from each presynaptic population.
(TIF)

**S4 Fig. Black flash estimates from LPA on simulated CSD with penalty on deviations from zero in CSD summed across channels.** (A) LPA-estimated black flash CSD generated from firing in each presynaptic population for $\lambda = 3$, the value at which the deviation from 0 in the CSD summed across channels was smallest (Fig 5A). (B) True black flash CSD generated from firing in presynaptic populations. (C) Relative MSE (left) and correlation (right) between CSD generated from each presynaptic population estimated by CSD and true CSD generated from each presynaptic population at different values of $\lambda$. LGN: green, V1: black, feedback: red, background: purple. (D) Relative MSE and correlation between LPA-estimated and true black flash CSD generated from each presynaptic population at $\lambda = 3$.
(TIF)

**S5 Fig. LPA on simulated CSD using temporal profiles estimated from MUA in decomposition and one population for V1.** (A) Top: LPA-estimated white flash CSD generated from firing in each presynaptic population. (B) True white flash CSD generated from firing in the same presynaptic populations. (C) Relative MSE and correlation between LPA-estimated and true white flash CSD generated from each presynaptic population. (D) Top: LPA-estimated black flash CSD generated from firing in each presynaptic population. (E) True black flash CSD generated from firing in the same presynaptic populations. (F) Relative MSE and correlation between LPA-estimated and true black flash CSD generated from each presynaptic population.
(TIF)

**S6 Fig. Penalizing deviations from zero in CSD summed across channels with presynaptic temporal profiles estimated from MUA.** (A) Average sum of LPA-estimated total CSD across channels (black line) with varying penalty ($\lambda$) on deviations from 0. Red dashed line: Average deviation from 0 of sum across channels for total simulated CSD to which LPA was applied. (B) Relative MSE (left) and correlation (right) between CSD from simulation and CSD estimated from LPA-components at different penalty magnitudes with temporal profile of V1 firing rates estimated from MUA. White flash: dark red line, black flash: blue line. (C) Total simulated CSD (top) and CSD estimated from LPA-components (bottom) in response to white and black full-field flash when the penalty for deviations from zero $\lambda = 4.5$, which is the value at which the deviation from zero for the average sum across channels in the LPA-estimated CSD is smallest (see (A)). (D) Relative MSE (left) and correlation (right) between LPA-

estimated and true white flash CSD generated from each presynaptic population at different values of λ. LGN: green, V1: black, feedback: red, background: purple. (E) Top: LPA-estimated white flash CSD generated from firing in each presynaptic population for λ = 4.5. Bottom: True white flash CSD generated from firing in presynaptic populations. (F) Relative MSE and correlation between LPA-estimated and true white flash CSD generated from each presynaptic population with λ = 4.5.
(TIF)

**S7 Fig. Avg. MUA power across depth in V1 model.** (A) The blue line shows the average MUA power at different channels of the simulated probe across all trials of white and black full-field flash stimuli.
(TIF)

**S8 Fig. MUA fit depends on number of recording channels in LGN.** (A) Relationship between number of recording channels in LGN in experiments and the correlation between the LPA-estimated and simulated MUA in LGN. Each blue star corresponds to an animal in the dataset and the red line displays the results of a linear regression between number of recording channels in LGN and the correlation.
(TIF)

**S9 Fig. ICA cannot identify the four major CSD contributors.** (A) Estimated white flash CSD of four ICA components. (B) True white flash CSD from each population. (C-D) Correlation and relative MSE, respectively, between CSD of ICA-components and true CSD in (A) and (B). (E) Estimated black flash CSD of four ICA components. (F) True black flash CSD from each population. (G-H) Correlation and relative MSE, respectively, between CSD of ICA-components and true CSD in (E) and (F).
(TIF)

**S10 Fig. PCA cannot identify the four major CSD contributors.** (A) Estimated white flash CSD of four PCA components. (B) True white flash CSD from each population. (C-D) Correlation and relative MSE, respectively, between CSD of PCA-components and true CSD in (A) and (B). (E) Estimated black flash CSD of four PCA components. (F) True black flash CSD from each population. (G-H) Correlation and relative MSE, respectively, between CSD of ICA-components and true CSD in (E) and (F).
(TIF)

**S11 Fig. Applying LPA to simulated data with two populations for V1 distinguished.** (A) Temporal profiles of firing rates of presynaptic excitatory populations in external structures (LGN: light green line; feedback: red line; background: purple line), upper layer V1 merged (blue line), and deep layers V1 merged (dark green line). (B) Top: Trial-averaged CSD from simulation with white (left) and (black) flash stimuli and CSD estimated from LPA-components using presynaptic firing rates in (A) in decomposition. Bottom: Relative MSE and correlation between CSD from simulation and CSD estimated from LPA-components. (C) Temporal profiles of CSD of each LPA-component in response to white and black flash. (D) Top: LPA-estimated white flash CSD generated from firing in each presynaptic population shown in (A). Bottom: True white flash CSD generated from firing in the same presynaptic populations. Right: Relative MSE and correlation between LPA-estimated and true white flash population CSD. (E) Top: LPA-estimated black flash CSD generated from firing in each presynaptic population shown in (A). Bottom: True black flash CSD generated from firing in the same presynaptic populations. Right: Relative MSE and correlation between LPA-estimated

and true black flash population CSD.
(TIF)

**S12 Fig. Results from applying LPA to all 24 animals in the Visual Coding dataset.**
(PDF)

## Author Contributions

**Conceptualization:** Atle E. Rimehaug, Anders M. Dale, Gaute T. Einevoll.

**Data curation:** Atle E. Rimehaug.

**Formal analysis:** Atle E. Rimehaug, Anton Arkhipov, Gaute T. Einevoll.

**Investigation:** Atle E. Rimehaug, Anders M. Dale, Anton Arkhipov, Gaute T. Einevoll.

**Methodology:** Atle E. Rimehaug, Anders M. Dale, Anton Arkhipov, Gaute T. Einevoll.

**Project administration:** Anders M. Dale, Gaute T. Einevoll.

**Resources:** Anton Arkhipov, Gaute T. Einevoll.

**Software:** Atle E. Rimehaug.

**Supervision:** Anders M. Dale, Anton Arkhipov, Gaute T. Einevoll.

**Validation:** Atle E. Rimehaug.

**Visualization:** Atle E. Rimehaug.

**Writing – original draft:** Atle E. Rimehaug.

**Writing – review & editing:** Atle E. Rimehaug, Anton Arkhipov, Gaute T. Einevoll.

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
