## [Decision Letter · Decision Letter 0]

2 May 2024

Dear Mr Rimehaug,

Thank you very much for submitting your manuscript "Uncovering population contributions to the extracellular potential in the mouse visual system using Laminar Population Analysis" for consideration at PLOS Computational Biology.

As with all papers reviewed by the journal, your manuscript was reviewed by members of the editorial board and by several independent reviewers. In light of the reviews (below this email), we would like to invite the resubmission of a significantly-revised version that takes into account the reviewers' comments.

Please address all comments and make sure to address in detail the important concerns by Reviewer 2 and the layer distinction issues brought up by the other two reviewers.

We cannot make any decision about publication until we have seen the revised manuscript and your response to the reviewers' comments. Your revised manuscript is also likely to be sent to reviewers for further evaluation.

Sincerely,

Hermann Cuntz

Academic Editor

PLOS Computational Biology

Thomas Serre

Section Editor

PLOS Computational Biology

Please address all comments and make sure to address in detail the important concerns by Reviewer 2 and the layer distinction issues brought up by the other two reviewers.

Reviewer's Responses to Questions

**Comments to the Authors:**

Reviewer #1: This is a very thorough manuscript, using advanced computational modeling and data analysis to investigate the biophysical properties of neural signals. The authors apply a method that they call “Laminar Population Analysis” (LPA) to disentangle and thereby better interpret the local field potential. For that, they use both simulated and recorded data from mouse primary visual cortex (V1). The work is clearly positioned in previous literature, the methods are rigorous and well described. The figures are clear, and the results are very carefully interpreted. The results are only partially successful, as stated by the authors : “The contributions from each laminar population could not be satisfactorily separated and identified with LPA.” This limited success is particularly the case for layer 4. I have to comments about this aspect of the work.

First, neurons in V1 are thought to be driven preferentially by oriented edges, while the only stimuli used in this study are full-field flashed. The activation with full field flashes can be expected to be simpler, not leading to a differential effect between layer 4 and other layers, compared to when using patterned stimuli such as gratings, as also shown in studies in primates (see Schroeder et al. 1991 Vision Research, https://doi.org/10.1016/0042-6989(91)90040-c). It would therefore be relevant to at least compare the results to Gabor patches, which would be possible using the dataset from the study in mice that was used for the current work (Siegle et al. 2021 Nature).

Second, layers in the mouse visual cortex are not so clearly differentiated as in primates, in particular lacking a clearly defined layer 4c (Lein et al. 2017; https://doi.org/10.1146/annurev-neuro-070815-013858; Jorstad, …, Lein 2023 https://doi.org/10.1126/science.adf6812). It would therefore be expected that LPA would work much better in primate V1. I can imagine that analyzing data from previous laminar studies from primate V1 would go beyond the aim of the current study. But it would be relevant to add that this difference between species could be a possible explanation for the lack of a the success to isolate layer 4 in mice.

Reviewer #2: The correct interpretation of LFP in vivo has been the focus of many debates and studies. The authors here present a novel approach aiming at disentangling the contribution of different populations "based on physiological rather than statistical assumptions", building on their previous work and in particular on the Laminar Population Analysis published in 2007.

The aim of the work is clear and so are the description of the methods and the illustration of the results, including the figures. Also a "minor" element of the work as the idea of penalizing deviations from zero in the summed CSD has definitely a general interest.

However, I think that there is a central point that might not have been properly assessed by the authors. One of the claims of the abstract is that the method is able to disentangle the contribution to LFP of local and remote inputs. But if I correctly understand, in order to do this the authors need to exploit the knowledge of the subcortical and remote cortical firing rates, which are available from the model selected as ground truth. Now, how can this be useful in experimental settings where these firing rates are typically not available?

This also puts another claim at stake. The authors state that assumptions underlying ICA and PCA decomposition of LFP are not always valid, and I agree, but then claim that their method is superior. But could it be otherwise, given that the LPA approach is provided with data from recordings that are not included in the ICA and PCA decomposition? Should'nt the comparison be performed starting from the same set of recordings?

There is one sentence in the limitation section that might refer to this - but seems more focusing on the unlikelihood of having access to all the local excitatory firing rates, which is indeed a limitation but less severe than the unlikelihood of knowing excitatory firing rates also from all projecting regions.

Could the author please clarify this key issue?

Reviewer #3: This study describes and evaluates Laminar Population Analysis, a decomposition algorithm which facilitates interpretation of extracellular signals, an important research area. The algorithm is evaluated on the Allen Brain V1 model and experimental data. The choice of model is very strong as it is highly detailed and has been thoroughly validated against experimental data (including recently LFP/CSD patterns in a recent publication). Similarly, the experimental datasets used are high quality and provide abundant data for statistical validation. The authors demonstrate the usefulness of the algorithm to 1) identify laminar positions and temporal profiles of population rates, and 2) estimate CSD sinks and sources generated by long-range inputs and V1 recurrent activity, outperforming previous PCA and ICA methods. Although the algorithm's performance separating the contribution of different V1 layers is limited, I believe this is an important negative result to report. It can lead to a better understanding of the system -- e.g. high synchrony is limiting this separation, as pointed out by the authors -- and sets a benchmark for future improved versions or other algorithms. The authors also demonstrate how to apply the algorithm to real V1 in vivo data, making this study substantially more impactful.

Below I list some concerns and suggestions to address before publication of the manuscript:

Abstract: It would be useful to mention in the abstract the negative results regarding estimation of generators in cortical layers, and maybe the high synchrony cause -- I understand there is limited space, but one sentence or less would probably be enough.

Line 32: a brief explanation of return currents might be useful for those not familiar with the topic

Line 173: "the correlations between the LPA-estimated and the simulated MUA" - I think this first results section could benefit from some more details of how the algorithm works and what it is trying to do. For example, here it might be confusing to readers why the algorithm is generating an estimate of the MUA if the MUA in itself is an input to the model; maybe clarify that it is trying to decompose the MUA into its components/populations.

Line 184: "whether the layer position of each recording electrode determined by LPA corresponds to the true layer position of that electrode" - similarly, here it is unclear how the algorithm is how the electrode location is determined by LPA; there is a lot of detail about the Metrics (which could be partly moved to Methods), but it is missing more intuitive insights into what LPA is doing / encoding

Line 251: would be useful to clarify if 1) the population temporal profiles include both excitatory and inhibitory neurons, and if so, 2) if they are multiplied by the same kernel given that E vs I would have opposite effects on the CSD ?

Fig 3 - some font sizes too small

Line 348: "did not improve" - this was surprising to me. Any intuition as to why firing rates provide a worse performance that MUA? intuitively it seems it should be the other way around?

Line 371: "We continued to use the firing rates" - at this point this choice was not clear since before you had shown worse performance.

Line 402: "increased from 0.46 to 0.6" - this is using the firing rates instead of the MUA; but comparing MUA-based results both the LGN and FB decreased significantly. I guess this explains the choice of firing rates method? This should all be clarified to avoid confusing the reader.

Line 465: It was great to see how the approach was iteratively improved. I was still surprised to see that using MUA the LGN result was very low (0.03) -- any explanation for this? Again these results justify the use of firing rates and not MUA. However, when applying to experimental data, one would need to use MUA? Should the MUA results be therefore reported instead of the firing rate ones? And since the MUA results are worse, does this reduce the applicability of the algorithm?

Line 568: This results subsection would benefit from a few final sentences clarifying the meaning of the results and any issues, as was done in previous subsections.

Line 626: It would be interesting to demonstrate that high synchrony is causing the issue, perhaps using a V1 simulation with different inputs, even if not realistic. However, this is just a suggestion and not strictly required for this revision, perhaps to be done in future work.

**Have the authors made all data and (if applicable) computational code underlying the findings in their manuscript fully available?**

Reviewer #1: Yes

Reviewer #2: Yes

Reviewer #3: None

PLOS authors have the option to publish the peer review history of their article (what does this mean?). If published, this will include your full peer review and any attached files.

Reviewer #1: No

Reviewer #2: No

Reviewer #3: No
---

## [Decision Letter · Decision Letter 1]

1 Oct 2024

Dear Dr Rimehaug,

Thank you very much for submitting your manuscript "Uncovering population contributions to the extracellular potential in the mouse visual system using Laminar Population Analysis" for consideration at PLOS Computational Biology. As with all papers reviewed by the journal, your manuscript was reviewed by members of the editorial board and by several independent reviewers. The reviewers appreciated the attention to an important topic. Based on the reviews, we are likely to accept this manuscript for publication, providing that you modify the manuscript according to the review recommendations.

Please do make ammends to the paper with respect to the criticism by the reviewers. This is a requirement for us to proceed with this paper.

Sincerely,

Hermann Cuntz

Academic Editor

PLOS Computational Biology

Thomas Serre

Section Editor

PLOS Computational Biology

Please do make ammends to the paper with respect to the criticism by the reviewers. This is a requirement for us to proceed with this paper.

Reviewer's Responses to Questions

**Comments to the Authors:**

Reviewer #1: I had two comments. The second was well answered by the authors, but the first was not. In response to my first comment, the authors wrote :

"We agree with the reviewer that patterned stimuli could potentially leadto greater differentiation in the response between different populations. However, with a single probe recording activity within a structure like V1, the LPA method is only suited for unpatterned stimuli. The reason for that is that for unpatterned stimuli, the spiking activity recorded from neurons near the probe can reasonably be assumed representative of the spiking activity of all neurons in the column. For patterned stimuli, this is unlikely to hold true because neurons close to the probe and those laterally further away are exposed to different stimulation patterns. In a multi-shank arrangement where spiking activity is recorded at multiple different locations in the column, patterned stimuli may also be appropriate for the application of the LPA method."

This point might be true for complicated natural stimuli. However, I suggested to use gratings, which are one dimensional stimuli and would therefore give very similar input to neighboring columns. One-dimensional gratings is what has been most commonly used to measure CSD responses in V1 (together with checkerboard stimuli, so 2-dimensional gratings). Finally, measuring the responses using either drifting gratings or Gabor patches would be possible using the dataset that was used for the current work (Siegle et al. 2021 Nature).

Reviewer #2: The answers provided to my comments by the authors are correct and complete. Still, no improvement was made in the Discussion or at least in the Methods to make these two crucial points clearer to the reader. I definitely recommend to modify the manuscript to include the clarifications proposed by authors in the reply.

Reviewer #3: All comments have been successfully addressed. Congratulations on this important work.

**Have the authors made all data and (if applicable) computational code underlying the findings in their manuscript fully available?**

Reviewer #1: Yes

Reviewer #2: Yes

Reviewer #3: None

PLOS authors have the option to publish the peer review history of their article (what does this mean?). If published, this will include your full peer review and any attached files.

Reviewer #1: No

Reviewer #2: No

Reviewer #3: **Yes: **Salvador Dura-Bernal

Figure Files:

Data Requirements:

Reproducibility:

References:

---

## [Editor Report · Decision Letter 2]

20 Nov 2024

Dear Dr Rimehaug,

We are pleased to inform you that your manuscript 'Uncovering population contributions to the extracellular potential in the mouse visual system using Laminar Population Analysis' has been provisionally accepted for publication in PLOS Computational Biology.

Best regards,

Thomas Serre

Section Editor

PLOS Computational Biology

Thomas Serre

Section Editor

PLOS Computational Biology

Feilim Mac Gabhann

Editor-in-Chief

PLOS Computational Biology

Jason Papin

Editor-in-Chief

PLOS Computational Biology

---

## [Editor Report · Acceptance letter]

30 Nov 2024

PCOMPBIOL-D-24-00081R2 

Uncovering population contributions to the extracellular potential in the mouse visual system using Laminar Population Analysis

Dear Dr Rimehaug,

I am pleased to inform you that your manuscript has been formally accepted for publication in PLOS Computational Biology. Your manuscript is now with our production department and you will be notified of the publication date in due course.

With kind regards,

Zsofia Freund
